# Struct2D: A Perception-Guided Framework for Spatial Reasoning in MLLMs

**Fangrui Zhu**[1,*] **Hanhui Wang**[1,3,*] **Yiming Xie**[1], **Jing Gu**[4], **Tianye Ding**[1]
**Jianwei Yang**[2], **Huaizu Jiang**[1]

[1] Northeastern University    [2] Microsoft Research
[3] University of Southern California [4] University of California, Santa Cruz
[1]{zhu.fang, wang.hanh, xie.yim, ding.tian, h.jiang}@northeastern.edu,
[2]jw2.yang@gmail.com, [3]hanhuiwa@usc.edu, [4]jgu110@ucsc.edu
https://github.com/neu-vi/struct2d

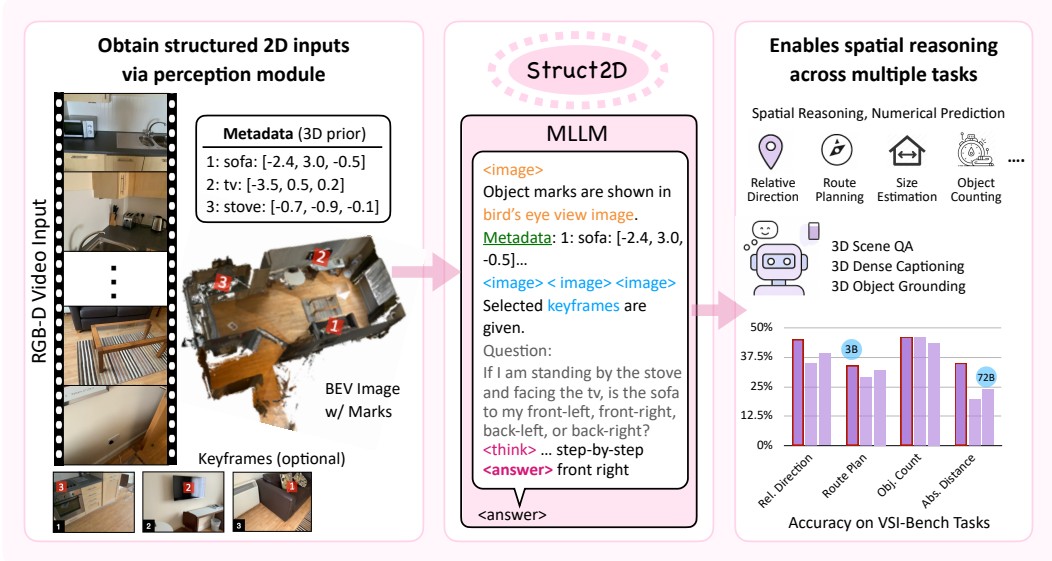

Figure 1: **Overview of our** `Struct2D` **framework for enabling spatial reasoning in Multimodal Large Language Models (MLLMs).** From an RGB-D video, we generate structured 2D inputs—BEV images with filtered object marks, object-centric metadata, and optional keyframes—via a 3D perception module. These inputs prompt an MLLM with spatial priors and visual context, enabling diverse spatial reasoning tasks without explicit 3D input at inference.

## Abstract

Unlocking spatial reasoning in Multimodal Large Language Models (MLLMs) is crucial for enabling intelligent interaction with 3D environments. While prior efforts often rely on explicit 3D inputs or specialized model architectures, we ask: *can MLLMs reason about 3D space using only structured 2D representations derived from perception?* We introduce **Struct2D**, a perception-guided prompting framework that combines bird's-eye-view (BEV) images with object marks and object-centric metadata, optionally incorporating egocentric keyframes when needed. Using Struct2D, we conduct an in-depth zero-shot analysis of closed-source MLLMs (*e.g.*, GPT-o3) and find that they exhibit surprisingly strong spatial reasoning abilities when provided with structured 2D inputs, effectively handling tasks such as relative direction estimation and route planning. Building

---

*Equal contribution.

on these insights, we construct **Struct2D-Set**, a large-scale instruction tuning dataset with 200K fine-grained QA pairs across eight spatial reasoning categories, generated automatically from 3D indoor scenes. We fine-tune an open-source MLLM (Qwen2.5VL) on Struct2D-Set, achieving competitive performance on multiple benchmarks, including 3D question answering, dense captioning, and object grounding. Our approach demonstrates that structured 2D inputs can effectively bridge perception and language reasoning in MLLMs-without requiring explicit 3D representations as input. We will release both our code and dataset to support future research.

# 1   Introduction

Understanding objects and their spatial relationships in 3D space is a cornerstone of intelligent interaction in complex physical environments. Tasks such as robotic manipulation [38, 67], autonomous navigation [25, 56], and visual reasoning [3, 10, 14, 53, 82, 101] all depend on accurate spatial understanding of scenes. At the core of these tasks lies the ability to localize objects precisely and reason about their configurations in 3D space. Moreover, grounding such spatial understanding in natural language enhances an AI system's ability to interpret, explain, and act upon spatial information in human-centric contexts.

Traditional task-specific models rely on explicit 3D representations as input, such as point clouds or reconstructed environments [3, 34, 53, 102], providing detailed geometric information. However, these models, often trained on limited data sources, making them less adaptable and struggle to generalize to diverse and complex textual queries. As a result, they fail to effectively bridge spatial reasoning with language comprehension, limiting their applicability for embodied AI.

In recent years, Multimodal Large Language Models (MLLMs) [24, 40, 60, 96] developed with Large Language Models (LLMs) have achieved significant advances in perception and reasoning tasks for images and videos. To extend MLLMs' capabilities to 3D understanding, point cloud-based LLMs [13, 23, 27, 28, 31, 54, 61, 62, 73, 79] have emerged, incorporating 3D spatial features by aligning point cloud data with LLMs. This integration enhances spatial reasoning and provides a richer understanding of the 3D physical world. However, they often rely on well-annotated datasets for instruction tuning and require point-cloud features as input, which limits their flexibility.

Unlike models that take explicit 3D representations as input, humans perceive the world as a continuous stream of 2D visual inputs akin to a *video*, and naturally infer spatial relationships and object configurations by building mental representations subconsciously [58, 70]. Naturally, we ask "*Can MLLMs perform spatial reasoning* without *using explicit 3D features as direct inputs?*" Recent work has begun to explore this direction by leveraging cognitive maps [82] and Bird's Eye View (BEV) images [63] generated from video as 2D spatial cues, enabling MLLMs to perform spatial reasoning [81, 85]. While promising, these approaches often omit object appearance and detailed priors (*e.g.*, coordinates, categories), which are critical for comprehensive 3D understanding.

We conduct an in-depth analysis of MLLMs' spatial reasoning abilities using a **perception-guided 2D framework** called `Struct2D` Prompting. This strategy transforms 3D perception outputs—obtained from off-the-shelf detectors—into structured 2D inputs, consisting of (1) a rendered bird's-eye-view (BEV) image with projected object marks[2] and (2) object-centric metadata such as category labels and 3D coordinates. When appearance cues are needed, we optionally incorporate egocentric keyframes selected based on object visibility. This design enables MLLMs to reason about complex 3D scenes using only structured 2D visual and textual cues, eliminating the need for explicit 3D inputs. We begin by evaluating this approach on GPT-o3 [60], a representative closed-source MLLM, to assess its zero-shot spatial reasoning capabilities.

To better understand the spatial reasoning capabilities of existing MLLMs, we begin with a zero-shot analysis using our proposed `Struct2D` Prompting strategy. The goal is to evaluate whether a pretrained, closed-source model such as GPT-o3 can accurately infer 3D spatial relationships when given only structured 2D visual and textual inputs. We use rendered bird's-eye-view (BEV) images with projected object marks and object-centric metadata, allowing the model to reason about 3D scenes without access to explicit 3D features. This analysis yields several key insights: (1) A single

---

[2]We follow the term "mark" as used in [81].

informative BEV image, combined with metadata, is often sufficient for accurate zero-shot 3D scene understanding; (2) Prompt composition is critical—different spatial reasoning tasks benefit from tailored input formats; (3) For challenging tasks in VSI-Bench [82], such as egocentric-to-allocentric transformations, MLLMs can perform robustly when provided with well-structured 2D projections of the 3D scene.

Guided by the findings from our zero-shot analysis, we construct a large-scale instructional tuning dataset, named `Struct2D-Set`, using an automated pipeline. The dataset consists of 200K QA pairs generated from 6K 3D indoor scenes, leveraging ground-truth object annotations provided by the original 3D datasets. It spans eight categories of spatial reasoning tasks relevant to embodied AI. To ensure data quality, we use ChatGPT to both enrich the QA pairs with step-by-step reasoning traces and identify potentially low-quality samples. Additionally, we incorporate a human-in-the-loop review process to further refine and validate the dataset. We then fine-tune an open-source MLLM (Qwen2.5VL [72]) using `Struct2D-Set`. Although the fine-tuned model is evaluated under noisy 3D perception conditions, it achieves strong performance across multiple spatial reasoning benchmarks, including 3D question answering [3, 53, 82], spatial captioning [14], and object grounding [10, 95], demonstrating the practicality and robustness of our approach.

Our main contributions are as follows:

- We propose a perception-guided 2D prompting strategy, `Struct2D` Prompting, and conduct a detailed zero-shot analysis that reveals MLLMs' ability to perform 3D spatial reasoning from structured 2D inputs alone.
- We introduce `Struct2D-Set`, a large-scale instructional tuning dataset with automatically generated, fine-grained QA pairs covering eight spatial reasoning categories grounded in 3D scenes.
- We fine-tune an open-source MLLM to achieve competitive performance across several spatial reasoning benchmarks, validating the real-world applicability of our framework.

## 2 Related Work

**3D Spatial Reasoning with MLLMs.** Developing real-world embodied agents requires equipping Multimodal Large Language Models (MLLMs) with robust 3D spatial reasoning abilities [8, 9, 16, 42, 48, 83, 99]. Recent efforts have explored spatial understanding through language [57, 65, 77], static 2D images [52, 55, 64, 69, 81], or videos [26, 45, 63, 82]. Our work builds upon the video-input setting, but diverges by enabling spatial reasoning in MLLMs using only structured 2D inputs—BEV images, object marks, and metadata—without relying on explicit 3D encoder / representations at inference.

**Instruction Tuning for 3D Spatial Reasoning.** Recent work [13, 35, 43, 44, 45] has explored instruction tuning to enhance MLLMs' capabilities for 3D spatial reasoning, targeting tasks such as 3D visual grounding [2, 10, 95], 3D dense captioning [14], and 3D question answering [3, 53]. M3DBench [43] provides region- and scene-level instruction-response pairs for general 3D understanding, while 3DMIT [44] focuses on scene-centric instructions. LL3DA [13] supports interactive planning and reasoning across omni-3D inputs. Robin3D [35] introduces a 3D LLM trained on diverse instruction-following examples. R1-Zero-VSI [45] proposes a video-based instruction tuning dataset and a GRPO-based training method to enhance spatial reasoning in Qwen2-VL, yet its QA pairs involve limited reasoning complexity, cover fewer task types, and yield marginal performance gains. In contrast, we propose `Struct2D-Set`, a large-scale dataset that enables open-source MLLMs to acquire rich 3D spatial reasoning skills through instruction tuning—using only structured 2D representations, without requiring direct access to 3D point clouds.

**3D Point Cloud LLMs.** Recent advances in 3D point cloud LLMs enable natural language generation and interaction grounded in 3D geometry by directly processing point clouds as input. These models benefit from the geometric precision and texture richness of point clouds, offering stronger spatial understanding than raw image or video inputs. Prior work has focused on object-level [27, 61, 62, 79] and scene-level [13, 23, 28, 31, 54, 73] spatial reasoning. However, directly using point cloud features requires additional training and infrastructure, limiting flexibility and scalability in real-world applications.

**Prompting LLMs.** Despite the rapid scaling of large language models (LLMs)[1, 5, 17, 22, 33, 71, 93], their reasoning capabilities remain heavily dependent on effective prompt design. In-context

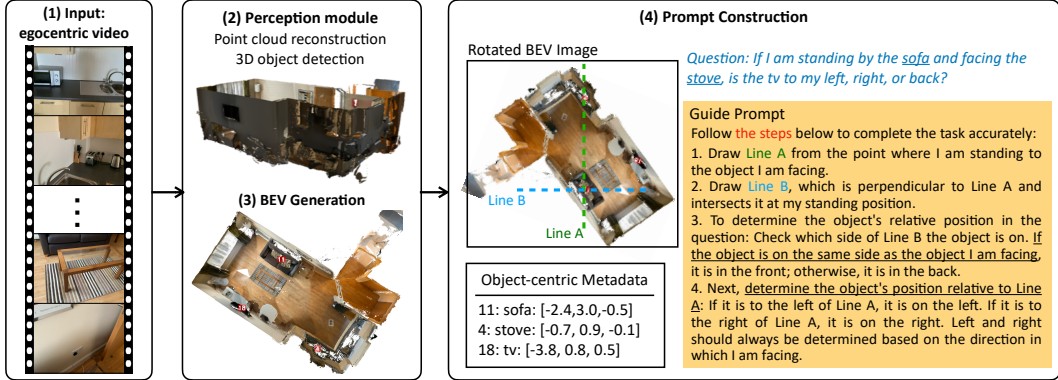

Figure 2: **Illustration of** `Struct2D` **prompting.** Given an egocentric video, we first reconstruct a point cloud and detect 3D objects. A bird's-eye-view (BEV) image is rendered and drawn with object marks related with the question. To facilitate reasoning about relative directions, the BEV is rotated to align with the agent's facing direction. We further construct object-centric metadata and a structured guide prompt to support the model in understanding spatial relationships between objects.

learning[5, 21], which conditions models on a few representative examples, has become a widely adopted technique for improving instruction-following behavior. To further enhance reasoning, strategies such as chain-of-thought [74] and tree-of-thought [86] prompting have been proposed. More recently, Multimodal Large Language Models (MLLMs)[15, 20, 24, 40, 41, 46, 50, 60, 80, 91, 92, 96, 100] have gained prominence for their ability to reason over multiple input modalities. This has led to a surge of research into prompting techniques tailored for MLLMs[7, 16, 29, 30, 39, 47, 49, 59, 68, 75, 76, 78, 81, 84, 85, 89, 94, 98]. Building on this direction, we propose `Struct2D`, a structured 2D prompting strategy that enables MLLMs to perform 3D spatial reasoning effectively—without requiring explicit 3D input representations.

## 3 Analysis on `Struct2D` prompting with GPT-o3

### 3.1 `Struct2D` Prompting

Given a video $\mathbf{V}$ as input, an MLLM $\mathcal{F}$ processes a set of $N$ sampled video frames, denoted as $\mathbf{I} = \{I_1, I_2, \ldots, I_N\}$, where each frame $I_n$ has dimensions $\mathbb{R}^{H \times W \times 3}$ for $n \in \{1, ..., N\}$. Alongside visual input, the MLLM receives a text query of length $l_i$, represented as $\mathbf{T}^{\text{in}} = [t_1^i, \ldots, t_{l_i}^i]$. The model then generates a textual response of length $l_o$, denoted as $\mathbf{T}^{\text{out}} = [t_1^o, \ldots, t_{l_o}^o]$, formulated as:

$$\mathbf{T}^{\text{out}} = \mathcal{F}(\mathbf{I}, \mathbf{T}^{\text{in}}). \tag{1}$$

However, directly using video frames for spatial reasoning introduces two major limitations: (1) **Incomplete perception** — Video frames are typically sampled sparsely and from limited viewpoints, which can result in missing critical visual evidence required for spatial understanding. For instance, consider a scene where a chair is tucked partially under a table. If most sampled frames are taken from frontal views or from a standing height, the chair's presence might be obscured or entirely invisible, leading the model to incorrectly assume there is empty space beneath the table. This limitation becomes more severe in cluttered or occluded environments, where small objects or those blocked by other furniture. (2) **Lack of global context** — Video frames offer fragmented, egocentric views that often fail to capture the overall structure of the scene. For example, determining whether a lamp is closer to the couch or the bookshelf may be impossible if the two objects never co-occur in the same frame. Without a consistent top-down or holistic representation, the model must rely on spatial memory or reasoning across disjoint perspectives—an ability that remains weak in most MLLMs. This fragmentation also impedes the understanding of traversability (*e.g.*, identifying a clear path from the door to the kitchen) or relational queries (*e.g.*, which chair is directly behind the dining table).

Table 1: **Zero-shot evaluation of GPT-o3 on the VSI-Bench subset.** The first row simply uses 16 frames from the input video, proposed in VSI-Bench [82]. For our prompting, we only input a BEV image with object marks on it along with object-centric meta information.

| Settings | # images | Cost ($) | Avg. | Numerical Answer | | | Multiple-Choice Answer | | |
|---|---|---|---|---|---|---|---|---|---|
| | | | | Obj. Count | Abs. Dist. | Room Size | Rel. Dist. | Rel. Dir. | Route Plan |
| VSI-Bench [82] | 16 | 105.07 | 48.6 | 44.3 | 34.1 | 50.9 | 51.0 | 49.4 | 61.9 |
| GPT4Scene [63] | 9 | 78.67 | 50.3 | 51.5 | 35.3 | **58.0** | 50.5 | 47.9 | 58.8 |
| Ours (Noisy Objects) | 1 | **27.25** | **56.1** | **52.8** | **38.4** | 48.9 | **60.0** | **60.1** | **76.2** |
| Ours (GT Objects) | 1 | **27.25** | 83.8 | 93.8 | 90.6 | 47.4 | 96.5 | 94.4 | 80.1 |

To address these issues, Struct2D incorporates a perception module $\phi_{\text{percept}}$ that extracts point clouds $\mathcal{P}$ and object detections $\mathcal{O}$ from the input video $\mathbf{V}$. We then generate a top-down bird's-eye-view image with filtered object marks—only including objects relevant to the question, as illustrated in Figure 2. Additionally, we construct object-centric metadata $\mathbf{T}^{\text{meta}}$ (*e.g.*, categories, coordinates) as textual input to guide spatial reasoning. Formally, we redefine Eq. 1 as:

$$\mathbf{T}^{\text{out}} = \mathcal{F}(\texttt{Struct2D}(\phi_{\text{percept}}(\mathbf{V}), \mathbf{T}^{\text{meta}}), \mathbf{T}^{\text{in}}). \tag{2}$$

For questions requiring appearance or depth cues (*e.g.*, object color or size), we supplement the BEV view with selected egocentric keyframes $\mathbf{I}_{\text{keyframe}}$ that capture clear views of the relevant objects. Instead of uniformly sampling keyframes, we use 3D projections to select frames that better capture the spatial coverage of the scene. The full formulation becomes:

$$\mathbf{T}^{\text{out}} = \mathcal{F}(\texttt{Struct2D}(\phi_{\text{percept}}(\mathbf{V}), \mathbf{T}^{\text{meta}}, \mathbf{I}_{\text{keyframe}}), \mathbf{T}^{\text{in}}). \tag{3}$$

This formulation illustrates how Struct2D leverages 3D perception as an intermediate step to generate informative 2D inputs that preserve spatial structure. Although 3D point clouds are used during preprocessing, they are not directly provided to the MLLM. Instead, they are transformed into BEV images and metadata used for prompting. As a result, the model performs spatial reasoning effectively without requiring explicit 3D representations as input.

**Evaluation Setup.** We sample questions from VSI-Bench [82], which is designed to evaluate complex spatial reasoning skills. Compared to traditional 3D QA datasets [3, 53], VSI-Bench covers more fine-grained object perception requirements, intricate global spatial relationships, and egocentric-to-allocentric transformations. It also features diverse indoor scene sources and robust evaluation metrics that go beyond rule-based NLP scoring.

**Comparison to GPT4Scene Prompting [63].** While GPT4Scene pioneered 2D spatial prompting using BEV images, our Struct2D strategy introduces several key improvements: (1) Filtered object marks tailored to the query improve visual relevance and reduce distraction; (2) Guided metadata prompts provide additional spatial priors; (3) Keyframe selection

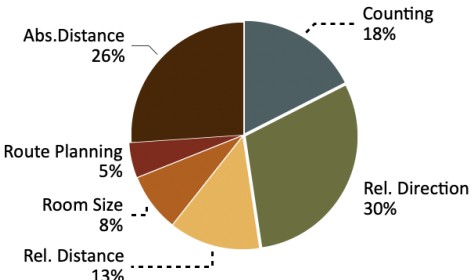

Figure 3: **Distribution of question types in the selected VSI-Bench subset.** This follows the distribution of the full set.

is optimized using depth-aware 3D projection instead of uniform sampling, making them both fewer and more informative (training drops from 6 to 4 hours).

### 3.2 Zero-shot Analysis of Struct2D Prompting

We construct a subset of 422 QA pairs for evaluation, selected due to API call budgets. As shown in Figure 3, the distribution of question types is consistent with the full benchmark. For our analysis, we generate object marks using both ground-truth 3D annotations and noisy detections (following [31, 63]), ensuring comprehensive object coverage while eliminating perception errors. This also enables a fair comparison with prior work, particularly [63].

**Zero-shot Prompting Results.** Table 1 shows that GPT-o3 exhibits strong spatial reasoning capabilities when prompted with structured 2D inputs. Specifically, providing both object-centric metadata

and filtered object marks significantly boosts performance, achieving 96.5 on relative distance, 94.4 on relative direction, and 80.1 on route planning. This highlights that explicit 3D representations are not strictly necessary—MLLMs can reason effectively with carefully structured 2D projections. The ablation further reveals that rotation alignment and a structured guide prompt each contribute to improved accuracy on relative direction tasks, with the combination of both yielding the best performance (94.4). These results underscore the importance of aligning spatial context and guiding the model through geometric reasoning steps. Notably, our method requires only a single BEV image and lightweight metadata, making it a low-cost and robust alternative to multi-frame prompting strategies [63, 82].

Table 2: **Ablation on different prompting strategies.**

| Metadata | Filtered Marks | Rel. Dist. | Rel. Dir. | Route Plan | | Guide Prompt | Rotation | Rel. Dir. |
|---|---|---|---|---|---|---|---|---|
| – | – | 67.5 | 82.1 | 74.3 | | – | – | 75.3 |
| – | ✓ | 72.1 | 88.3 | 78.3 | | – | ✓ | 89.2 |
| ✓ | – | 75.3 | 89.5 | 50.6 | | ✓ | – | 80.2 |
| ✓ | ✓ | **96.5** | **94.4** | **80.1** | | ✓ | ✓ | **94.4** |

(a) **Effects of metadata and filtered marks.**   (b) **Effects of rotation and guide prompt.**

**What makes a good prompt for spatial reasoning?** Table 2 highlights the impact of key components in our prompting strategy. Incorporating object-centric metadata consistently improves performance across tasks—raising relative distance accuracy from 67.5 to 96.5 and route planning from 74.3 to 80.1—highlighting its importance for grounding spatial context. Filtering object marks based on question relevance further reduces ambiguity, yielding substantial gains in route planning (from 50.6 to 80.1). For relative direction, both the use of a structured guide prompt and rotation alignment prove essential. While each individually improves accuracy (89.2 and 80.2 respectively), their combination leads to the best performance (94.4). We focus on these question types in ablation because they represent core challenges in spatial understanding.

## 4 Large-Scale Instruction Tuning with `Struct2D-Set`

Building on the insights from our zero-shot analysis (Sec. 3), we construct a large-scale instruction tuning dataset, `Struct2D-Set`, tailored to support diverse spatial reasoning tasks grounded in realistic 3D indoor scenes. Notably, the dataset is designed to require only 2D projected inputs during training, enabling efficient supervision without reliance on full 3D data.

In this section, we first describe the design and construction of `Struct2D-Set`, highlighting its coverage, annotation pipeline, and task diversity. We then present the supervised fine-tuning (SFT) setup using open-source MLLMs, detailing the model configurations and training procedures. Finally, we evaluate the effectiveness of our instruction-tuned model across multiple spatial reasoning benchmarks, assessing its generalization and reasoning capabilities.

### 4.1 `Struct2D-Set`

**Overview.** `Struct2D-Set` consists of 200K QA pairs generated from over 6K richly annotated indoor scenes, sourced from large-scale 3D reconstruction datasets—ARKitScenes[4], ScanNet [19], and ScanNet++[88]. These datasets capture diverse real-world environments, including homes, offices, and industrial settings. The QA pairs cover eight categories of spatial reasoning tasks. Figure 4 shows the distribution of question types across the dataset.

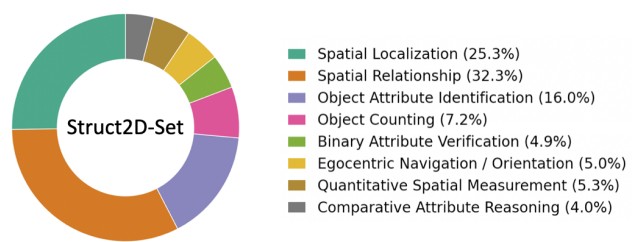

Figure 4: **Distribution of QA types in `Struct2D-Set`.** The dataset covers a diverse range of spatial reasoning skills, with a focus on spatial relationships and localization tasks that require strong geometric understanding.

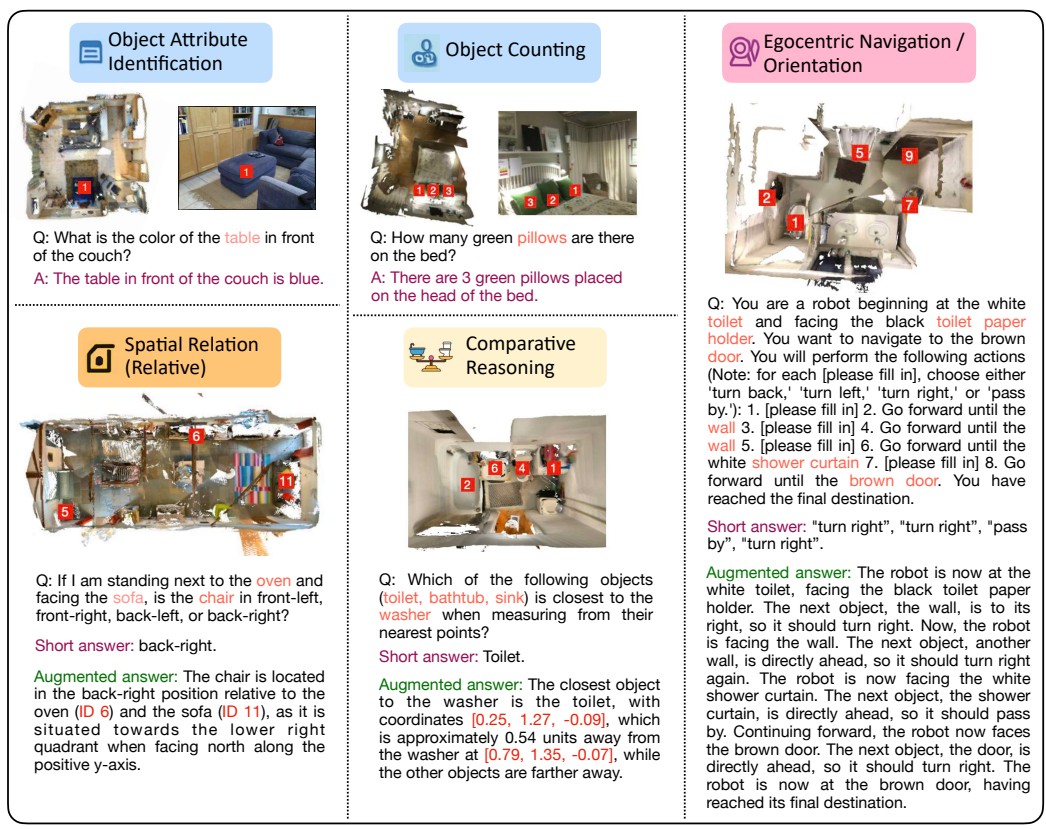

Figure 5: **QA examples of** `Struct2D-Set`. Examples cover diverse spatial reasoning tasks, including object attributes, counting, relative positioning, navigation, and comparative reasoning. Each QA pair includes a short answer from 3D geometry and an augmented answer with detailed reasoning generated by ChatGPT.

**Construction pipeline.** We generate two types of QA pairs to support both spatial reasoning and scene understanding tasks. Each type encompasses multiple subtypes targeting distinct reasoning skills. Representative examples from both types are shown in Figure 5.

The first type, inspired by VSI-Bench [82], involves tasks that require understanding global spatial relationships in 3D, such as spatial relation identification, egocentric navigation, and comparative reasoning. These questions cannot be answered from a single keyframe alone. We begin by extracting ground-truth object annotations from the training sets of the 3D datasets, including object boxes, depth maps, and camera poses. Using structured templates, we generate initial QA pairs based on this meta information, and then enrich them using ChatGPT to produce step-by-step reasoning traces and more natural language formulations. Each QA pair includes a short answer derived from geometry templates and a long-form answer elaborating on the reasoning process.

The second type of QA pairs is adapted from existing 3D scene understanding benchmarks, including ScanQA [3], SQA3D [53], Scan2Cap [14], ScanRefer [10], and Multi3DRefer [95]. These examples cover tasks such as object attribute identification, counting, and binary verification. We augment the original training set questions and descriptions using ChatGPT to improve clarity and reasoning depth. These tasks typically benefit from selecting keyframes where relevant objects are clearly visible, allowing the model to ground spatial reasoning in egocentric frames.

## 4.2 Experiment Setup

We fine-tune the open-source MLLM Qwen2.5VL [72] using our proposed dataset, `Struct2D-Set`. For evaluation, we primarily focus on VSI-Bench [82], which includes complex spatial reasoning tasks

Table 3: **Performance comparison of various models on VSI-Bench [82].** The model fine-tuned with `Struct2D-Set` surpasses both the `Struct2D` prompting and the video-based tuning baseline.

| Methods | Avg. | Numerical Answer | | | | Multiple-Choice Answer | | |
|---|---|---|---|---|---|---|---|---|
| | | Obj. Count | Abs. Dist. | Room Size | Obj. Size | Rel. Dist. | Rel. Dir. | Route Plan |
| *Open-source Models* | | | | | | | | |
| InternVL2-2B [15] | 30.3 | 21.8 | 24.9 | 35.0 | 22.0 | 33.8 | 44.2 | 30.5 |
| InternVL2-8B [15] | 33.9 | 23.1 | 28.7 | 39.8 | 48.2 | 36.7 | 30.7 | 29.9 |
| LongVILA-8B [80] | 21.1 | 29.1 | 9.1 | 0.0 | 16.7 | 29.6 | 30.7 | 32.5 |
| VILA-1.5-8B [46] | 29.5 | 17.4 | 21.8 | 18.8 | 50.3 | 32.1 | 34.8 | 31.0 |
| LongVA-7B [91] | 31.1 | 38.0 | 16.6 | 22.2 | 38.9 | 33.1 | 43.3 | 25.4 |
| LLaVA-NeXT-Video-7B [96] | 36.3 | **48.5** | 14.0 | 24.2 | 47.8 | **43.5** | 42.4 | 34.0 |
| LLaVA-OneVision-0.5B [40] | 31.2 | 46.1 | 28.4 | 28.3 | 15.4 | 28.9 | 36.9 | 34.5 |
| LLaVA-OneVision-7B [40] | 33.5 | 47.7 | 20.2 | 12.3 | 47.4 | 42.5 | 35.2 | 29.4 |
| R1-Zero-VSI [45] (Qwen2-VL-7B) | 32.1 | 39.4 | 25.0 | 43.2 | 25.8 | 32.6 | 30.9 | 27.8 |
| R1-Zero-VSI [45] (Qwen2-VL-7B) + SFT | 38.8 | 44.7 | 27.6 | **50.4** | 46.1 | 34.0 | 35.7 | 33.0 |
| *Ours* | | | | | | | | |
| Qwen2.5-VL-3B | 25.6 | 27.0 | 22.0 | 25.6 | 32.5 | 17.5 | 28.9 | 25.6 |
| Qwen2.5-VL-3B (`Struct2D Prompting`) | 29.4 | 46.6 | 24.6 | 22.3 | 33.6 | 21.2 | 30.5 | 27.2 |
| Qwen2.5-VL-3B (Baseline) | 33.9 | 24.6 | 34.0 | 46.4 | 53.5 | 21.2 | 30.5 | 27.2 |
| Qwen2.5-VL-3B (SFT) | 41.9 | 46.0 | 34.7 | 42.6 | 56.4 | 35.1 | 44.9 | 33.5 |
| Qwen2.5-VL-7B (SFT) | **43.6** | 47.1 | **35.1** | 48.9 | **57.1** | 35.1 | **45.9** | **35.8** |

such as relative direction and route planning. Additionally, we assess model performance on three standard 3D scene understanding tasks built on ScanNet [19]: 3D question answering (ScanQA [3], SQA3D [53]), 3D dense captioning (Scan2Cap [14]), and 3D visual grounding (ScanRefer [10], Multi3DRef [95]). For VSI-Bench, we input only BEV images with filtered object marks and metadata, as the tasks focus purely on spatial relationships. For the other benchmarks, which often involve object attributes or visual details, we additionally provide selected egocentric keyframes to support reasoning.

## 4.3 Implementation Details

We adopt Qwen2.5VL [72] as our base MLLM for instruction tuning. During training, the model receives BEV images with filtered object marks and object-centric metadata. For tasks that require appearance or attribute information (e.g., object color or count), we additionally provide egocentric keyframes. All visual inputs are resized to $480 \times 480$, and object marks are adaptively scaled based on their original resolution.

For questions involving complex spatial reasoning, such as relative direction or route planning, we insert special tokens `<think>` and `</think>` to guide the model to generate a step-by-step reasoning process, followed by the final answer enclosed within `<answer>` and `</answer>`. For simpler questions involving object appearance or quantitative estimation, the model is trained to directly produce short answers without reasoning traces. We train the model for one epoch using a base learning rate of 2e-6 with cosine annealing. Training with the whole `Struct2D-Set` takes approximately 8 hours on 8×H200 GPUs. For evaluation, we follow [31, 63] by reconstructing point clouds offline using BundleFusion [18], detecting 3D object boxes with Mask3D and UniDet, and projecting them into BEV images and 2D object marks.

## 4.4 Main results

We present quantitative results on VSI-Bench[82] in Table 3 and on ScanQA[3] and SQA3 [53] in Table 4. Additional benchmark results are provided in the Appendix due to space limitations.

As shown in Table 3, our model fine-tuned with the `Struct2D-Set` dataset achieves the highest average score (43.6) among all open-source models evaluated on VSI-Bench. Notably, it surpasses both the `Struct2D` prompting variant (29.4) and the standard baseline trained with uniformly sampled 16 video frames (33.9), confirming the effectiveness of our full instruction tuning approach. The performance gains are especially prominent on spatial reasoning tasks like relative direction (45.9) and route planning (35.8), where the model must integrate both geometric understanding and egocentric context. Compared with R1-Zero-VSI [45] (38.8), a recent method that trains Qwen2-VL-7B using video-based supervision, our tuned model not only achieves stronger average performance but also uses fewer visual frames and does not rely on dense temporal input. These results highlight the

Table 4: **3D Question Answering Evaluation on ScanQA [3] and SQA3D [53] datasets.**

| Methods | ScanQA(val) | | | | | SQA3D(val) | |
| --- | --- | --- | --- | --- | --- | --- | --- |
| | BLEU-1 | BLEU-4 | METEOR | ROUGE | CIDEr | EM-1 | EM-R1 |
| *Task-Specific Model* | | | | | | | |
| ScanQA [3] | 30.2 | 10.1 | 13.1 | 33.3 | 64.9 | – | – |
| SQ3D [53] | – | – | – | – | – | 46.6 | – |
| 3D-VLP [34] | 30.5 | 11.2 | 13.5 | 34.5 | – | – | – |
| 3D-Vista [102] | – | – | 13.9 | 35.7 | – | 48.5 | – |
| *3D LLM Based Model* | | | | | | | |
| Chat-3D [73] | 29.1 | 6.4 | 11.9 | 28.5 | 53.2 | – | – |
| Chat-3D v2 [31] | 38.4 | 7.3 | 16.1 | 40.1 | 77.1 | – | – |
| 3D-LLM [28] | 39.3 | 12.0 | 14.5 | 37.3 | 69.4 | – | – |
| LL3DA [13] | – | 13.5 | 15.9 | 37.3 | 76.8 | – | – |
| PQ3D [103] | – | – | – | – | – | 47.1 | – |
| LEO [32] | – | 11.5 | 16.2 | 39.3 | 80.0 | 50.0 | 50.0 |
| Chat-Scene [31] | 43.2 | 14.3 | 18.0 | 41.6 | 87.7 | 54.6 | 57.5 |
| *Vision LLM Based Model* | | | | | | | |
| InternVL-2-8B [15] | 23.9 | 3.3 | 14.5 | 34.3 | 62.5 | 33.0 | 45.3 |
| MiniCPM-V-2.6 [87] | 25.1 | 8.4 | 11.8 | 31.5 | 60.1 | 42.6 | 46.6 |
| Qwen2-VL-7B (GPT4Scene) | 43.4 | 14.6 | **17.7** | 43.6 | 90.9 | 57.4 | 60.7 |
| Qwen2.5-VL-7B (Ours) | **45.2** | **15.8** | 17.4 | **44.1** | **92.1** | **58.5** | **61.3** |

Table 5: **Ablation on different variants.** To save computational resource, models are trained with Qwen2.5VL-3B model by default.

| Settings | Avg. | Numerical Answer | | | Multiple-Choice Answer | | |
| --- | --- | --- | --- | --- | --- | --- | --- |
| | | Obj. Count | Abs. Dist. | Room Size | Rel. Dist. | Rel. Dir. | Route Plan |
| *Tuning Data Format* | | | | | | | |
| wo/ augmented QA | 31.5 | 43.7 | 33.1 | 34.1 | 32.1 | 14.7 | 31.5 |
| w/ augmented QA | 38.0 | 44.4 | 33.6 | 41.5 | 33.3 | 42.2 | 33.0 |
| *Evaluation Strategy* | | | | | | | |
| wo/ `</think>` | 36.2 | 44.1 | 33.6 | 41.5 | 33.3 | 38.6 | 26.3 |
| w/ `</think>` | 36.1 | 44.4 | 30.0 | 35.6 | 31.5 | 42.2 | 33.0 |

scalability and efficiency of `Struct2D-Set` for training capable spatial reasoners without explicit 3D features at inference.

Table 4 shows results on two traditional 3D question answering benchmarks, ScanQA and SQA3D. Our model outperforms most existing methods, including several that rely on explicit 3D point cloud inputs. Compared with GPT4Scene [63], our model performs on par across most metrics. However, these benchmarks primarily require identifying relevant keyframes and generating free-form textual answers. As a result, models can often rely on memorizing object-level attributes, and the rule-based evaluation metrics (*e.g.*, BLEU, CIDEr) may not fully reflect the correctness or reasoning quality of the generated answers. Please refer to Appendix for qualitative results. **Ablation Study** To better understand the impact of individual components in our framework, we conduct a series of ablation studies using the Qwen2.5VL-3B model for efficiency, as shown in Table 5. First, we evaluate the effect of QA augmentation. Incorporating enriched QA pairs generated with ChatGPT leads to a substantial improvement in overall performance (Avg: 38.0 vs. 31.5), especially on reasoning-heavy tasks such as relative direction (42.2 vs. 14.7). This supports our earlier claim that step-by-step reasoning traces help guide the model's attention and inference. We further assess the role of explicit reasoning supervision using the `<think>` and `<answer>` tokens. While the average scores are comparable, including `<think>` tokens improves performance on reasoning-intensive tasks like relative direction and route planning (42.2 vs. 38.6 and 33.0 vs. 26.3, respectively), indicating that instructing the model to explicitly reason can enhance output quality on complex spatial questions. Figure 6 illustrates step-by-step reasoning generated using our `<think>` and `<answer>` prompting format in egocentric navigation tasks. In each case, the model is prompted to analyze object positions and spatial transitions from a top-down BEV image with object markers. These examples highlight the model's ability to decompose spatial reasoning tasks into interpretable steps and generate structured answers grounded in visual context.

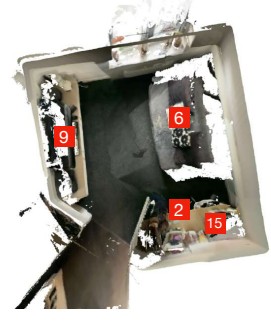

Here are object labels and ids: table:2, sofa:6, tv:9, table:15 in the bird's eye view image. If I am standing by the tv and facing the sofa, is the table to my front-left, front-right, back-left, or back-right? Here are the options: ['A. front-right', 'B. front-left', 'C. back-left', 'D. back-right']. Please think step by step and enclose your reasoning in <think> </think> tags and then provide the short answer with only the option letter in <answer> </answer>.

☆ Model's reasoning steps:

<think>The table, identified as marker ID 2, is located in the front-right quadrant relative to my position by the TV (marker ID 9) while facing the sofa (marker ID 6). </think>

✓ Final prediction: <answer> A </answer>

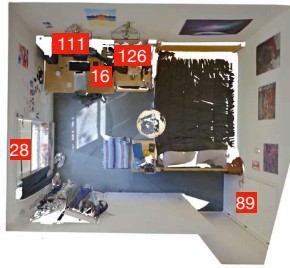

Here are object labels and ids: window:28, monitor:111, door:89, monitor:126, monitor:16 in the bird's eye view image. You are a robot beginning at the door and facing the window. You want to navigate to the monitor. You will perform the following actions (Note: for each [please fill in], choose either 'turn back,' 'turn left,' or 'turn right.'): 1. Go forward until the heater 2. [please fill in] 3. Go forward until the monitor. You have reached the final destination. Here are the options: ['A. Turn Left', 'B. Turn Back', 'C. Turn Right']. Please think step by step and enclose your reasoning in <think> </think> tags and then provide the short answer with only the option letter in <answer> </answer>.

☆ Model's reasoning steps:

<think> The robot is now at the door[89], facing the window [28]. The next object, the heater, is straight ahead, so it should go forward. Upon reaching the heater, the robot is still facing the window. The next object, the monitor, is to its right, so it should turn right. Now, the robot is facing the monitor. It goes forward until it reaches the monitor, which is the final destination. </think>

✓ Final prediction: <answer> A </answer>

Figure 6: **Example reasoning traces using `<think>` and `<answer>` prompting.** The top example shows relative direction reasoning, where the model infers the spatial relation between objects from a fixed viewpoint in the BEV image. The bottom example shows step-by-step planning, requiring the model to simulate orientation changes across a sequence of waypoints.

## 5 Limitations and future work

While `Struct2D` demonstrates strong spatial reasoning capabilities with structured 2D inputs, there remain areas where future work could extend its applicability:

- **3D preprocessing requirements.** Although `Struct2D` does not use 3D features during inference, it currently relies on 3D perception modules to generate BEV images and object-centric metadata. This may pose a challenge in latency-sensitive or resource-constrained environments. However, since `Struct2D` is agnostic to the specific perception backbone, it can readily integrate with ongoing advances in real-time and lightweight 3D reconstruction systems.
- **Indoor scene focus.** The current version of `Struct2D-Set` is constructed from over 6K richly annotated indoor scenes, including homes, offices, and classrooms. While this enables detailed reasoning in structured environments, generalization to outdoor or open-world scenes remains less explored. Incorporating diverse spatial layouts and object categories from outdoor domains is a promising direction for future dataset expansion.

## 6 Conclusion

We present `Struct2D`, a perception-guided framework that enables MLLMs to perform 3D spatial reasoning using structured 2D inputs. Through zero-shot analysis and instruction tuning, we show that BEV images, object-centric metadata, and keyframes are sufficient to unlock strong spatial reasoning capabilities—without requiring explicit 3D inputs. Our curated dataset, `Struct2D-Set`, supports scalable instruction tuning with fine-grained QA pairs grounded in real 3D scenes. Fine-tuning with

`Struct2D-Set` yields significant gains across spatial reasoning benchmarks, outperforming prior open-source methods even under noisy perception. These findings demonstrate that structured 2D projections are a practical and effective alternative to direct 3D representations, offering a scalable path toward robust multimodal spatial understanding in MLLMs.

## Acknowledgment

We thank Zhangyang Qi for the thoughtful discussions.

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

# A  Details of `Struct2D` Prompting Strategy.

Figure 7 illustrates the overall `Struct2D` prompting framework, which transforms egocentric 3D scene input into structured 2D representations for spatial reasoning. Given an input video and a spatial question, we first reconstruct a 3D point cloud from RGB-D frames and remove the ceiling to obtain a clear top-down view of the scene. Object detection is then performed in 3D space, and detected objects are projected onto a bird's-eye-view (BEV) image to produce a layout of the environment. These object marks are filtered to retain only those relevant to the input question.

We optionally extract egocentric keyframes to capture detailed object appearances. Keyframes are selected by projecting 3D object bounding boxes onto sampled video frames and depth maps, and identifying views where each object is both visible and unobstructed. Object-centric metadata—including object categories and 3D coordinates—is encoded as text and used as part of the prompt input.

Algorithm 1 outlines the core procedure for constructing the `Struct2D` prompt. Given an input video $\mathbf{V}$, depth frames $\mathbf{D}$, a reconstructed 3D scene $\mathcal{P}$, and a set of target objects $\mathcal{O}$, we begin by rendering a BEV image $v$ and projecting each object $o_i \in \mathcal{O}$ into the view using the RGB camera parameters $C_{\text{rgb}}$. The 2D projections are then drawn as object marks on the image.

To select keyframes, we sample $N$ RGB-D frames and iteratively check for visibility of objects not yet covered in the BEV. For each candidate frame $I_i$, we project the remaining unseen objects onto both the frame and its depth map. If a valid projection exists (i.e., the projected location lies within the image and has valid depth), the object mark is rendered and the frame is added to the keyframe set $\mathcal{I}_{\text{keys}}$. This process continues until all relevant objects are covered. The final prompt consists of ❶ a BEV image with filtered marks, ❷ optional keyframes containing visible objects, and ❸ object metadata text, all of which are passed to a multimodal large language model for reasoning.

This framework allows the MLLM to perform 3D spatial reasoning from 2D visual and textual inputs, without requiring direct access to raw 3D data at inference time. It enables scalable, flexible spatial understanding grounded in realistic perception outputs.

---

**Algorithm 1** `Struct2D` Visual Prompting

---

**Input:** Input video $\mathbf{V}$, Depth frames $\mathbf{D}$, Reconstructed 3D scene $\mathcal{P}$, Objects of interest $\mathcal{O}$, RGB camera parameters $\mathbf{C}_{\text{rgb}}$, Depth camera parameters $\mathbf{C}_{\text{d}}$

1: Render a Bird's Eye View image: $\mathbf{v} \leftarrow \text{BEV}(\mathcal{P})$
2: **for** $o_i \in \mathcal{O}$ **do**
3:     Project $o_i$ onto $\mathbf{v}$: $p_i \leftarrow \text{Project}(o_i, \mathbf{v}, \mathbf{C}_{\text{rgb}})$
4:     Update view: $\mathbf{v} \leftarrow \text{Add-Mark}(\mathbf{v}, p_i)$
5: **end for**
6: Sample $N$ frames: $\mathbf{I}, \mathbf{D_I} \leftarrow \text{Sample}(\mathbf{V}, \mathbf{D})$
7: Initialize key frame set: $\mathbf{I}_{\text{keys}} \leftarrow \{\}$
8: Initialize found objects set: $\mathcal{O}_{\text{F}} \leftarrow \{\}$
9: **for** $\mathbf{I_i} \in \mathbf{I}$ and $\mathbf{D_i} \in \mathbf{D_I}$ **do**
10:     $b_i \leftarrow \text{False}$
11:     **for** $o_j \in \mathcal{O}$ and $\notin \mathcal{O}_{\text{F}}$ **do**
12:         Project $o_j$ onto $\mathbf{I_i}$ and $\mathbf{D_i}$: $p_j^I \leftarrow \text{Project}(o_j, \mathbf{I_i}, \mathbf{C}_{\text{rgb}})$, $p_j^D \leftarrow \text{Project}(o_j, \mathbf{D_i}, \mathbf{C}_{\text{d}})$
13:         **if** $p_j^I \in \mathbf{I_i}$ and $p_j^D \in \mathbf{D_i}$ and $p_j^D \geq 0$ **then**
14:             $b_i \leftarrow \text{True}$
15:             Update frame: $\mathbf{I_i} \leftarrow \text{Add-Mark}(\mathbf{I_i}, p_j^I)$
16:             Add object to set: $\mathcal{O}_{\text{F}} \leftarrow \mathcal{O}_{\text{F}} \cup \{o_i\}$
17:         **end if**
18:     **end for**
19:     **if** $b_i$ **then**
20:         Add to key frame set: $\mathbf{I}_{\text{keys}} \leftarrow \mathbf{I}_{\text{keys}} \cup \{\mathbf{I_i}\}$
21:     **end if**
22: **end for**
**Return:** Informative BEV view $\mathbf{v}$ and key frame set $\mathbf{I}_{\text{keys}}$

---

**Qualitative comparison of our `Struct2D` prompting.** To better understand the impact of prompt design on spatial reasoning, we conduct qualitative analyses highlighting two key aspects of our

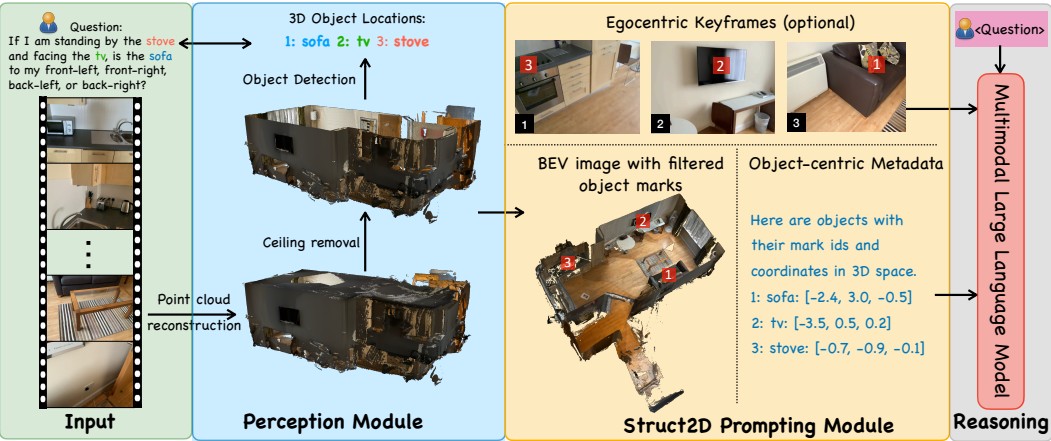

Figure 7: **Overview of the** `Struct2D` **Prompting Framework.** Given an egocentric video and a spatial question, we first reconstruct a 3D point cloud and remove the ceiling for a clear top-down view. Objects are detected in 3D space, and a bird's-eye-view (BEV) image is rendered with object marks projected onto the floor plane. These object marks are filtered based on the content of the question. We also extract egocentric keyframes by projecting 3D bounding box centers onto the video, when appearance cues are needed. Object-centric metadata—including object IDs and 3D coordinates—is encoded as text. The structured 2D visual and textual inputs are then fed into a multimodal large language model for spatial reasoning.

framework: reasoning guidance, object orientation, and structured metadata. As shown in Figure 8, when the model is prompted only with a BEV image and object marks, it struggles to accurately resolve relative spatial relationships. Adding a structured guide prompt enables the model to decompose the task into interpretable geometric steps, though it may still fail without an aligned reference frame. Once the BEV is rotation-aligned with the agent's viewpoint, the reasoning becomes more intuitive, leading to the correct answer. Similarly, in Figure 9, we illustrate the benefit of object-centric metadata. Without access to precise coordinates, the model must estimate distances visually, which can lead to errors. When provided with 3D object positions, the model can directly compute spatial relations such as Euclidean distances, significantly improving its accuracy on localization tasks. These examples highlight how prompt structure—through guided reasoning and geometric priors—plays a crucial role in unlocking spatial understanding in MLLMs.

# B    Details of `Struct2D-Set`

**Overview.** `Struct2D-Set` is a large-scale instruction tuning dataset aimed at enabling spatial reasoning and scene understanding in indoor 3D environments using only 2D projected inputs. It contains over 200K question-answer (QA) pairs derived from 6K richly annotated indoor scenes drawn from ScanNet [19], ScanNet++[88], and ARKitScenes[4]. Each QA instance is paired with structured scene- and object-level metadata, allowing models to learn spatial concepts without relying on explicit 3D feature representations during training.

The dataset spans eight categories of spatial reasoning tasks, such as object attribute identification, relative localization, and egocentric navigation. Each QA pair follows an instruction-style format and includes:

- A natural language question,
- A concise short-form answer,
- A long-form answer, when applicable, containing step-by-step reasoning or contextual elaboration,

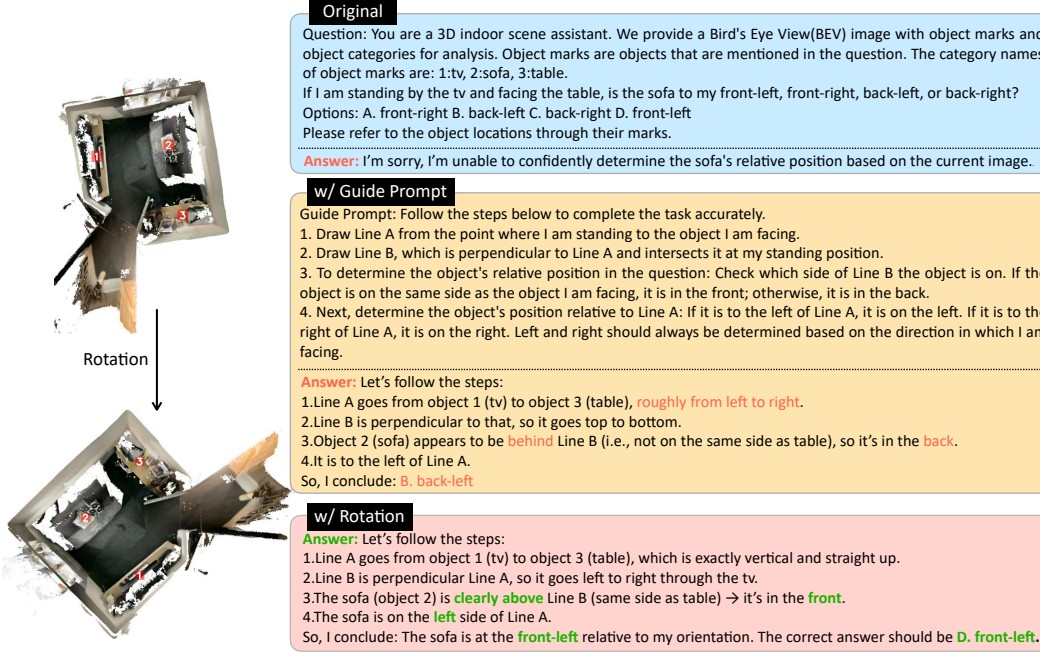

**Original**

Question: You are a 3D indoor scene assistant. We provide a Bird's Eye View(BEV) image with object marks and object categories for analysis. Object marks are objects that are mentioned in the question. The category names of object marks are: 1:tv, 2:sofa, 3:table.
If I am standing by the tv and facing the table, is the sofa to my front-left, front-right, back-left, or back-right?
Options: A. front-right B. back-left C. back-right D. front-left
Please refer to the object locations through their marks.

Answer: I'm sorry, I'm unable to confidently determine the sofa's relative position based on the current image.

**w/ Guide Prompt**

Guide Prompt: Follow the steps below to complete the task accurately.
1. Draw Line A from the point where I am standing to the object I am facing.
2. Draw Line B, which is perpendicular to Line A and intersects it at my standing position.
3. To determine the object's relative position in the question: Check which side of Line B the object is on. If the object is on the same side as the object I am facing, it is in the front; otherwise, it is in the back.
4. Next, determine the object's position relative to Line A: If it is to the left of Line A, it is on the left. If it is to the right of Line A, it is on the right. Left and right should always be determined based on the direction in which I am facing.

Answer: Let's follow the steps:
1.Line A goes from object 1 (tv) to object 3 (table), roughly from left to right.
2.Line B is perpendicular to that, so it goes top to bottom.
3.Object 2 (sofa) appears to be behind Line B (i.e., not on the same side as table), so it's in the back.
4.It is to the left of Line A.
So, I conclude: B. back-left

**w/ Rotation**

Answer: Let's follow the steps:
1.Line A goes from object 1 (tv) to object 3 (table), which is exactly vertical and straight up.
2.Line B is perpendicular Line A, so it goes left to right through the tv.
3.The sofa (object 2) is clearly above Line B (same side as table) → it's in the front.
4.The sofa is on the left side of Line A.
So, I conclude: The sofa is at the front-left relative to my orientation. The correct answer should be D. front-left.

Figure 8: **Effects of guide prompting and rotation correction in relative spatial reasoning of the zero-shot prompting.** We show different variations of our Struct2D. The original model struggles to interpret spatial relationships using the raw BEV image and object marks alone, failing to answer correctly. With the addition of a structured guide prompt, the model is able to decompose the reasoning process into geometric steps, though it still arrives at an incorrect answer. When object orientation is further corrected via rotation alignment, the spatial reference frame becomes more intuitive, and the model's reasoning becomes clearer and more accurate. This demonstrates the importance of both prompting design and canonical alignment in unlocking MLLMs' spatial understanding abilities. (Red texts are wrong answers; Green texts are correct ones.)

- Accompanying metadata including relevant object marks, spatial coordinates, and references to visual input modalities (*e.g.*, BEV image, selected keyframes).

Long-form answers are provided selectively for tasks that benefit from explicit reasoning or contextual understanding. For categories requiring direct factual responses—such as object counting or binary verification—only short-form answers are used. This balanced design ensures effective supervision across tasks of varying complexity, while maintaining interpretability and richness in reasoning. We next describe the construction process for each task category in detail.

**Object counting.** To construct object counting questions, we begin by sampling a scene from the training split of the source datasets and extracting its ground-truth object annotations. A target object category (*e.g.*, *chair*) is randomly selected from the annotated instances within the scene. A QA pair is then generated using a templated prompt such as *"How many class label(s) are there in this room?"*, paired with the correct numerical count as the answer. To improve linguistic diversity and fluency, we further augment these questions by prompting ChatGPT to generate alternative phrasings with equivalent semantic meaning.

**Spatial Relationship.** This category evaluates a model's ability to reason about the directional relationships between objects in a 3D scene from an egocentric perspective. Following the formulation in VSI-Bench [82], we focus on the subtask of *relative direction*, where the goal is to identify the directional location of a target object based on a specified standing point and facing direction.

To construct each QA pair, we begin by computing the 3D centers of all objects in the scene and projecting them onto the 2D BEV image. We then sample a triplet of objects representing the roles of *standing*, *facing*, and *target*, while filtering out ambiguous categories (*e.g.*, object clusters or large connected instances) and enforcing a minimum pixel-distance threshold to ensure spatial distinguishability. The <*standing*, *facing*> vector defines the forward direction of the agent, and the

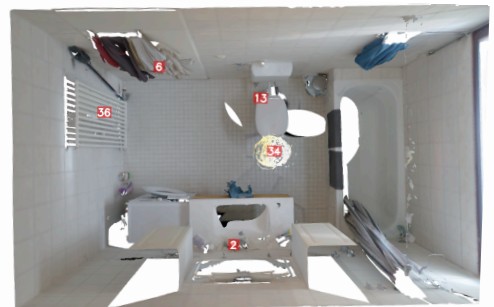

| Original | w/ Object-centric Metadata |
|---|---|
| Question: You are a 3D indoor scene assistant. We provide a Bird's Eye View(BEV) image with object marks and object categories for analysis. Object marks are objects that are mentioned in the question.
The category names of object marks are: 2: cup, 6: door, 13: toilet, 34: ceiling light, 36: heater.
Measuring from the closest point of each object, which of these objects (heater, cup, door, ceiling light) is the closest to the toilet? Options: A.Heater B. cup C. door D. ceiling light
Please refer to the object locations through their marks. | The center coordinates of the objects in the 3D point cloud are 2: cup: [1.45, 0.20, 0.98], 6: door: [0.69, 1.88, 1.06], 13: toilet: [1.68, 1.69, 0.49], 34: ceiling light: [1.69, 1.11, 2.44], 36: heater: [0.15, 1.44, 0.95]. |
| **Answer:** Let's analyze the relative distances in the BEV image:
**Toilet (13)** is near the upper center.
**Ceiling light (34)** is directly above or just slightly in front of the toilet.
**Cup (2)** is placed under the sink, somewhat farther toward the bottom of the image.
**Heater (36)** is on the far left wall.
**Door (6)** is on the top-left, implying it's on the floor plane.
Now compare the **closest point** of each object to the toilet:
•The **ceiling light (34)** is almost touching or directly above the toilet — essentially zero horizontal distance.
•The **cup (2)** is in front and below — still clearly farther.
•The **heater (36)** is across the room — farther.
•The **door (6)** is on the top — not spatially adjacent.
Correct Answer: D. ceiling light | **Answer:** We are given the center coordinates of the objects and asked to find which one is closest to the toilet (object 13). We approximate this by computing the Euclidean distance between the toilet and each of the other objects:
**Coordinates:**
•Toilet (13): [1.68, 1.69, 0.49]
•Cup (2): [1.45, 0.20, 0.98]
•Door (6): [0.69, 1.88, 1.06]
•Ceiling light (34): [1.69, 1.11, 2.44]
•Heater (36): [0.15, 1.44, 0.95]
**Distances to the toilet (using 3D Euclidean distance):**
•Cup: $\sqrt{(1.68-1.45)^2 + (1.69-0.20)^2 + (0.49-0.98)^2} \approx 1.584$
•Door: $\sqrt{(1.68-0.69)^2 + (1.69-1.88)^2 + (0.49-1.06)^2} \approx 1.158$
•Ceiling light: $\sqrt{(1.68-1.69)^2 + (1.69-1.11)^2 + (0.49-2.44)^2} \approx 2.034$
•Heater: $\sqrt{(1.68-0.15)^2 + (1.69-1.44)^2 + (0.49-0.95)^2} \approx 1.616$
**Conclusion:**
The **door** is the closest to the toilet.
**Answer: C. door** |

Figure 9: **Effect of object-centric metadata for precise spatial reasoning.** Originally, the model attempts to estimate distances based solely on the spatial layout in the BEV image but fails to identify the correct object closest to the toilet. In contrast, with access to object-centric metadata—specifically, 3D coordinates of each object—the model can compute accurate Euclidean distances and correctly identify the nearest object. This example highlights how structured metadata enhances geometric reasoning and helps avoid ambiguity in visual interpretation. (Red text indicates incorrect reasoning; Green text indicates the correct answer.)

<*standing*, *target*> vector is used to determine the relative orientation of the target object. The angular offset between these vectors is then discretized into directional bins such as *front-left*, *right*, or *back*, producing the correct label.

We format each QA pair using a natural language template (*e.g.*, *"If I am standing by the TV and facing the refrigerator, is the sink to my left, right, or back?"*) and provide the short-form directional answer. To enhance both linguistic variation and model supervision, we further augment each instance using ChatGPT, which paraphrases the question and generates a long-form answer that walks through the step-by-step reasoning process under the egocentric frame of reference.

**Comparative Reasoning.** This category involves tasks where the model must compare spatial attributes among multiple objects. We focus on *relative distance comparison*, where the objective is to identify which candidate object is closest or farthest from a given reference object.

To construct such questions, we first select a reference object whose identity is unambiguous based on its class label. Next, we sample a set of candidate objects, including multiple instances—potentially of the same class—to encourage instance-level discrimination. In contrast to reference selection, we do not filter ambiguous or repeated categories among the candidates, as the goal is to challenge the model to reason over instance-specific spatial relations.

We compute the 3D centroid of each object using the center of its oriented bounding box and measure pairwise Euclidean distances between the reference and each candidate. Based on the ranking of these distances, we generate a templated question, such as *"Measuring from the closest point of each object, which of these objects (candidate labels) is closest to the reference object?"*, along with the correct answer derived from the computed rankings.

To enhance linguistic variation and encourage deeper reasoning, we further augment each instance using ChatGPT, which paraphrases the question and generates a long-form answer. These enriched responses guide the model through comparative spatial reasoning before producing the final answer.

**Quantitative Spatial Measuring.** This category targets tasks requiring the model to reason about metric properties in 3D space, such as object size, spatial extent, and inter-object distance. We focus on the *object absolute distance* subtask, where the model needs to estimate the physical distance between two specified objects within a scene.

To construct these questions, we begin by selecting two distinct objects with clearly identifiable class labels to avoid semantic ambiguity. Using the oriented bounding box annotations, we extract all eight corner points for each object and compute the minimum Euclidean distance across all point pairs—this serves as the ground-truth physical distance between the two objects. Based on this calculation, we generate templated questions such as: *"Measuring from the closest point of each object, what is the distance between the object1 and the object2 (in meters)?"*

To enhance supervision and promote reasoning transparency, we further use ChatGPT to produce long-form answers. These responses walk through the spatial computation process, prompting the model to conceptually simulate pairwise distance comparisons before arriving at the correct numerical answer.

**Egocentric Navigation.** This category focuses on tasks that require the model to plan navigation routes from an egocentric perspective, reasoning about object references, turning actions, and scene layout. The goal is to simulate how an embodied agent would traverse a 3D space by following instructions grounded in object-level references.

To construct these tasks, we first sample up to 15 candidate objects per scene and project their 3D centers onto the BEV image. Each object is visually marked in the BEV, and a mark-to-label dictionary string is generated to facilitate object identification. These scene representations are then passed to ChatGPT to generate plausible navigation routes in natural language.

Route generation is guided by several constraints: ❶ Each route must consist of a sequence of consecutive object marks (IDs) that an agent can follow. ❷ At each step, the agent must perform a local navigation action (e.g., turn left, turn right, pass by). ❸ Routes must avoid collisions with irrelevant or obstructing objects. ❹ Each path should span 3 to 5 objects to ensure sufficient reasoning complexity.

All generated routes undergo human review to ensure spatial plausibility. Invalid routes are discarded, and valid ones are further augmented via route reversal and sub-segmentation to increase diversity.

To determine the action sequence along the path, we randomly choose a facing object at the starting point to establish the initial egocentric orientation. For each transition between objects, we compute the vector from the current object to the next and compare it with the current facing direction to infer the correct action (e.g., turn left, go forward). These navigation actions form the short-form answer.

For each object along the route, we apply our keyframe selection algorithm to extract egocentric views from the original video. These keyframes, combined with the object labels, are used to prompt ChatGPT to generate rich textual descriptions of each waypoint. Finally, we instruct ChatGPT to produce long-form answers that walk through the full navigation route, step by step, reasoning over orientation shifts and identifying the appropriate navigation action at each stage.

**Other Categories.** The remaining task types—such as object attribute identification and binary attribute verification—are constructed by augmenting QA pairs from existing 3D vision-language datasets, including ScanQA [3], SQA3D [53], Scan2Cap [14], ScanRefer [10], and Multi3DRefer [95]. These datasets provide scene-specific questions grounded in the ScanNet environment and collectively cover all eight spatial reasoning categories defined in `Struct2D-Set`.

To adapt these examples for instruction tuning, we first use ChatGPT to rephrase each question into a more natural and instructional style. For tasks requiring reasoning, we also prompt ChatGPT to generate long-form answers that walk through the inference process. For visual grounding, we localize referenced objects using two approaches: when object IDs are available, we apply our keyframe detection method to extract representative egocentric views. In datasets with descriptive references (e.g., SQA3D), we extract referring expressions with ChatGPT and apply Grounding DINO [51] and SAM [36] to identify and segment the mentioned objects.

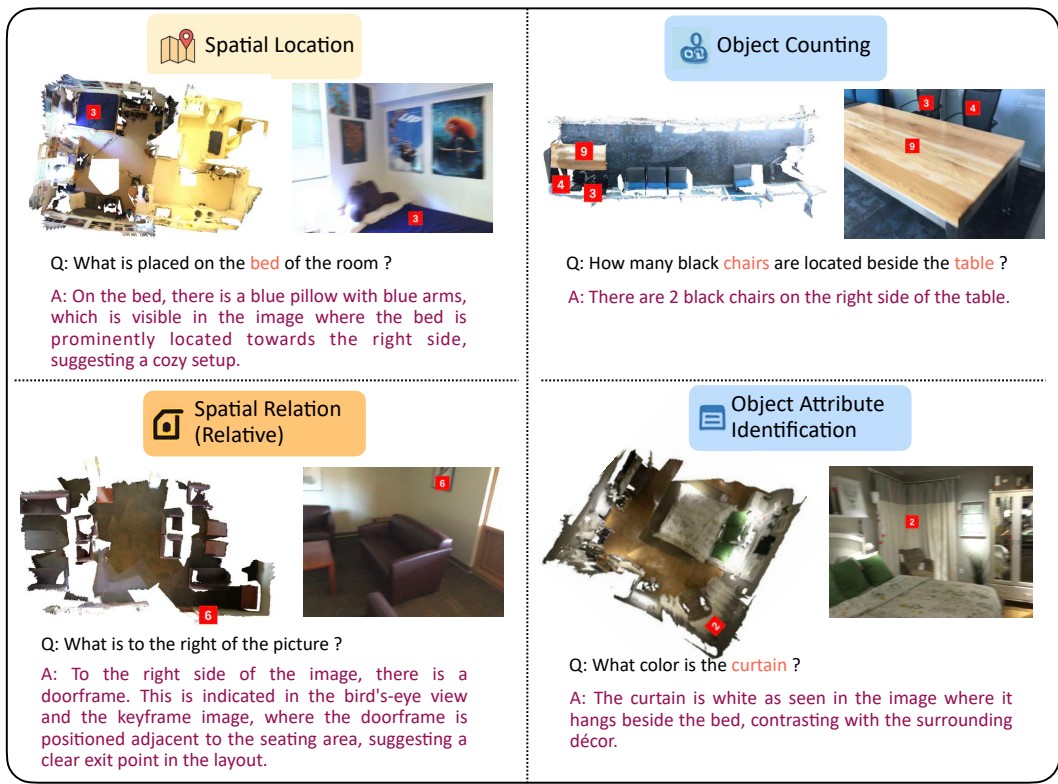

Figure 10: **Examples of QA pairs from `Struct2D-Set` used for instruction tuning.** Each example illustrates a distinct category of spatial reasoning: spatial localization, object counting, spatial relationship, and object attribute identification. For each question, the model is provided with a BEV image annotated with object marks, and optionally an egocentric keyframe to enhance visual grounding. The answers include descriptive reasoning grounded in object positions and appearances, enabling the model to learn to associate structured 2D inputs with fine-grained spatial understanding.

The resulting keyframes are paired with each QA instance to serve as visual inputs during fine-tuning. This pipeline enables instruction tuning on complex, object-centric spatial tasks while relying only on 2D visual projections and avoiding the need for explicit 3D geometry at training time.

## C    Implementation Details

We use Qwen2.5VL [72] as the base multimodal large language model (MLLM) for instruction tuning. During training, the model receives BEV images with filtered object marks and object-centric metadata as core inputs. For tasks requiring visual cues such as object color or quantity, we additionally provide egocentric keyframes. The BEV images are resized to $640 \times 640$, with object marks adaptively scaled based on their original image resolution. Keyframes are resized to $256 \times 246$ and stitched into compact 1×2 or 2×4 grids, enabling efficient batch loading and reducing GPU memory consumption.

To support reasoning supervision, we adopt a task-specific output format. For complex spatial reasoning tasks—such as relative direction estimation or route planning—we wrap the reasoning process between special tokens `<think>` and `</think>`, followed by the final answer enclosed within `<answer>` and `</answer>`. For tasks focused on appearance or simple measurements, the model is trained to generate direct short-form answers without explicit reasoning traces. The model is trained for one epoch using a base learning rate of 2e-6 with cosine annealing, taking approximately 8 hours on 8×H200 GPUs.

At evaluation time, we follow standard practices from prior work [31, 63], reconstructing point clouds offline using BundleFusion [18], detecting 3D objects using Mask3D[66] and UniDet [37],

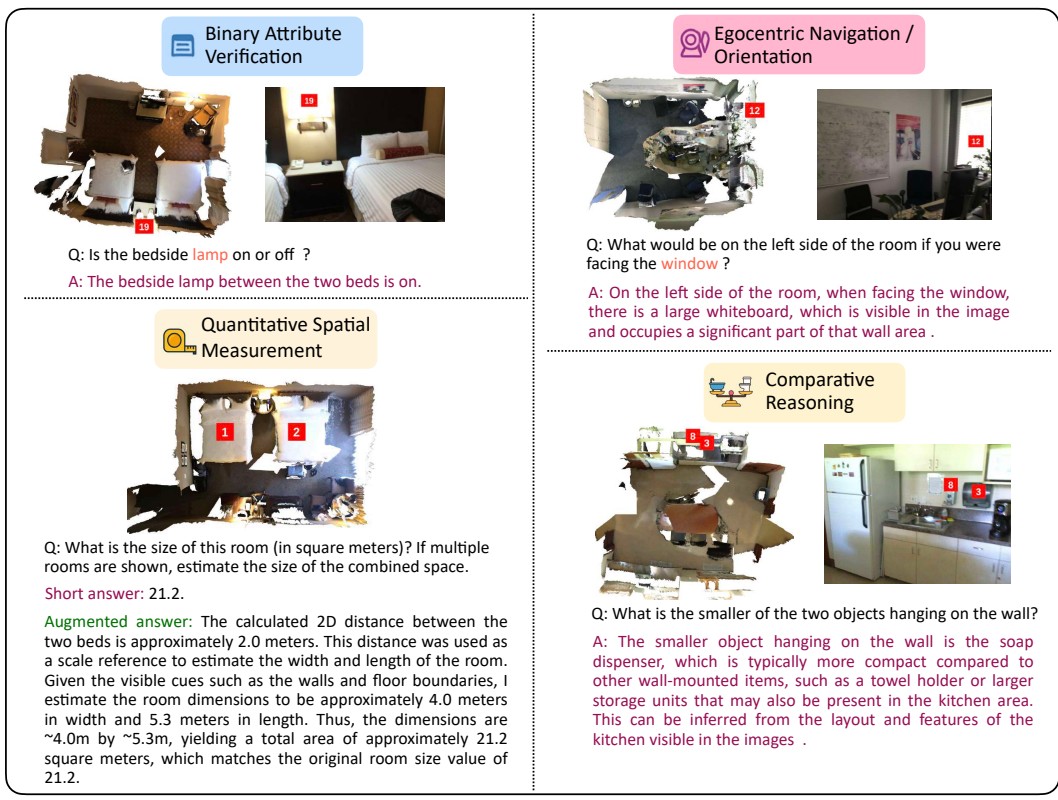

Figure 11: **Additional QA examples from** `Struct2D-Set` **covering diverse spatial reasoning categories.** This figure showcases examples from the remaining categories in our dataset: binary attribute verification, egocentric navigation and orientation, quantitative spatial measurement, and comparative reasoning. Each QA pair is grounded in structured 2D visual inputs (BEV views and keyframes) and enriched with object marks and contextual metadata. These examples demonstrate the model's ability to reason about object states, egocentric spatial references, metric estimations, and relative comparisons—key competencies for embodied spatial understanding.

and projecting the results to produce BEV images and 2D object marks. For object-level grounding, we apply a rule-based method to identify the relevant objects mentioned in each question.

## D    Results on 3D Grounding and 3D Dense Captioning

**Quantitative results.** Tables 6 and 7 present our model's performance on 3D grounding (ScanRefer, Multi3DRefer) and dense captioning (Scan2Cap) benchmarks. While our method does not achieve the highest scores under rule-based metrics such as all F1@0.25/0.5 and BLEU/ROUGE, it consistently delivers competitive results compared to existing vision-language baselines. Importantly, our approach does not rely on point cloud features during training or evaluation, in contrast to task-specific and 3D LLM models that depend heavily on explicit 3D representations. In addition, our approach requires substantially fewer egocentric keyframes on average (2 compared to 8 in GPT4Scene [63]), resulting in a more efficient and scalable training process. Compared to models designed for narrow tasks, our framework is more general and supports a wider range of spatial reasoning types, including relative direction and route planning, which are not covered by these benchmarks. It is also worth noting that the current evaluation metrics are rule-based and limited in expressiveness, which may not fully reflect a model's capability in spatial understanding.

**Qualitative results.** Figure 12 illustrates qualitative examples of our fine-tuned Qwen2.5-VL-7B model across three major spatial reasoning tasks: 3D dense captioning, object grounding, and 3D question answering. In each case, the model receives a BEV image with object marks, optionally supplemented with egocentric keyframes and metadata, and produces either a descriptive caption, an

Table 6: **3D Grounding Evaluation on ScanRefer [10] and Multi3DRefer [95] datasets.**

| Methods | ScanRefer (val) | | Multi3DRefer (val) | |
|---|---|---|---|---|
| | Acc@0.25 | Acc@0.50 | all F1@0.25 | all F1@0.50 |
| *Task-Specific Model* | | | | |
| 3DVG-Transformer [97] | 47.6 | 34.7 | – | 25.5 |
| 3DJCG [6] | 49.6 | 37.3 | – | 26.6 |
| D3Net [11] | – | 37.9 | – | 32.2 |
| M3DRef-CLIP [95] | 51.9 | 44.7 | 42.8 | 38.4 |
| *3D LLM Based Model* | | | | |
| Chat-Scene [31] | 55.5 | 50.2 | 57.1 | 52.4 |
| *Vision LLM Based Model* | | | | |
| Qwen2-VL-7B [72] | 5.4 | 5.1 | 21.1 | 19.9 |
| Qwen2-VL-7B (GPT4Scene [63]) | 40.5 | 36.7 | 45.4 | 42.1 |
| Qwen2.5-VL-7B (Ours) | 51.7 | 48.5 | 42.1 | 40.6 |

Table 7: **3D Dense Captioning Evaluation on Scan2Cap [14] dataset.**

| Methods | IoU@0.25 | | IoU@0.5 | |
|---|---|---|---|---|
| | BLEU-4 | ROUGE | BLEU-4 | ROUGE |
| *Task-Specific Model* | | | | |
| Scan2Cap [14] | 34.2 | 55.3 | 23.3 | 44.5 |
| 3DJCG [6] | 40.2 | 59.2 | 31.0 | 50.8 |
| X-Trans2Cap [90] | 35.7 | 54.7 | 25.1 | 45.3 |
| 3D-VisTA [102] | 36.5 | 57.6 | 34.0 | 54.3 |
| Vote2Cap-DETR [12] | 39.3 | 59.3 | 34.5 | 54.4 |
| *3D LLM Based Model* | | | | |
| LL3DA [13] | 41.4 | 59.5 | 36.8 | 55.1 |
| LEO [32] | – | – | 36.9 | 57.8 |
| Chat-Scene [31] | 38.2 | 60.6 | 36.3 | 58.1 |
| Robin3D [35] | – | – | 38.4 | – |
| *Vision LLM Based Model* | | | | |
| Qwen2-VL-7B [72] | 3.8 | 24.7 | 3.8 | 24.6 |
| Qwen2-VL-7B (GPT4Scene [63]) | 36.3 | 57.6 | 34.2 | 55.2 |
| Qwen2.5-VL-7B (Ours) | 34.8 | 57.0 | 32.7 | 54.5 |

object ID, or a short-form answer. The examples demonstrate the model's ability to reason about visual attributes (*e.g.*, "a brown rectangle"), relative spatial positions (*e.g.*, "the table is to the right of the couch"), and numerical or commonsense questions. We observe that the model often produces answers consistent with the ground truth, and in some cases offers additional descriptive clarity grounded in the visual context. These results highlight the effectiveness of our `Struct2D` prompting strategy in enabling rich spatial understanding from structured 2D inputs.

# E   Failure cases

To better understand the limitations of our approach, we conducted a qualitative error analysis on 30 representative questions spanning multiple QA types in VSI-Bench. Among the 16 failure cases, we identified two dominant causes. First, in 11 cases, the underlying 3D reconstruction was noisy or incomplete, producing degraded BEV projections that obscured critical spatial layouts. Second, in 5 cases, missing detections—often involving small or heavily occluded objects—led to incomplete structured inputs. Both factors reduce Struct2D's ability to encode accurate spatial cues, thereby hindering its reasoning capability. Representative visualizations of these failure modes are shown in

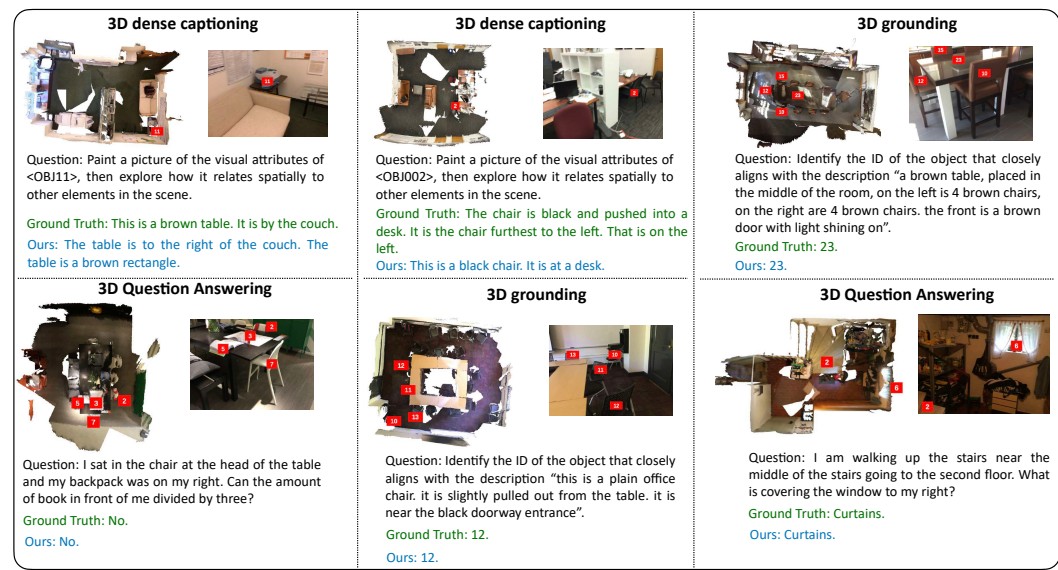

Figure 12: **Output examples from our fine-tuned Qwen2.5-VL-7B model across multiple 3D spatial reasoning tasks.** The figure showcases model responses on 3D dense captioning, object grounding, and 3D question answering tasks. Each example includes the question, BEV and keyframe inputs with object marks, the ground-truth answer, and our model's prediction. These examples illustrate the model's ability to localize, describe, and reason about spatial relations using structured 2D prompts derived from 3D scenes. Across tasks, the model demonstrates strong alignment with ground-truth answers, even when questions require appearance attributes, relative spatial context, or numerical reasoning.

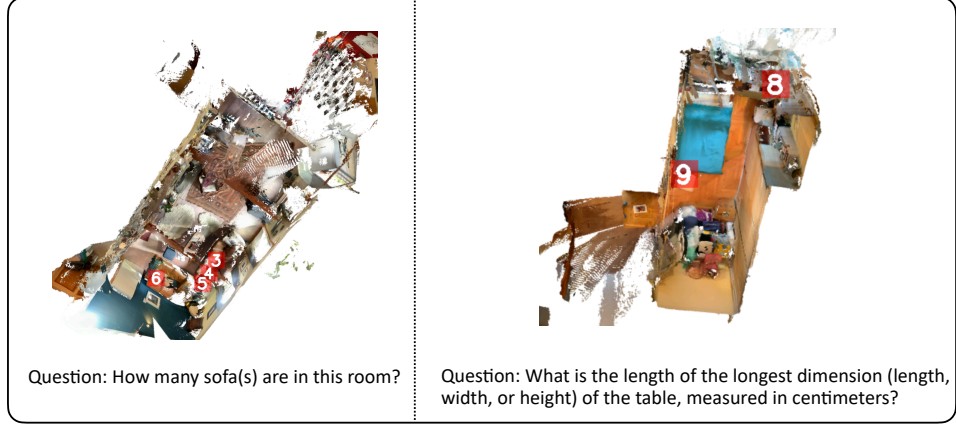

Figure 13: Examples of failure cases caused by 3D reconstruction and detection.

Figure 13. These examples highlight the importance of robust 3D reconstruction and reliable object grounding for spatial reasoning in complex indoor scenes.

# F    Broader impacts

Our work introduces `Struct2D`, a perception-guided prompting framework that enables Multimodal Large Language Models (MLLMs) to perform robust spatial reasoning in 3D environments using only structured 2D and text inputs. This direction offers several broader implications for research, society, and the reasonable development of AI systems.

**Social Benefits.** `Struct2D` lowers the barrier to 3D spatial reasoning by leveraging RGB-D perception instead of requiring dense 3D annotations or point cloud inputs during inference. This makes spatial understanding more accessible to a wide range of applications, especially in settings where real-time 3D sensing is noisy, sparse, or unavailable. Potential downstream applications include:

- **Assistive robotics**, where spatial-language understanding is critical for navigation and object manipulation in dynamic home environments;
- **Augmented reality interfaces**, where natural-language spatial queries must be resolved in partially reconstructed environments;
- **Accessibility technologies**, especially for users with visual impairments, by enabling robust, language-driven scene understanding with minimal hardware.

**Potential Negative Impacts.** The preprocessing pipeline relies on egocentric video and 3D reconstruction, which may involve scenes from private homes or workplaces. If deployed in real-world applications, such systems may inadvertently capture sensitive spatial or personal data. Ensuring strict anonymization, access control, and user consent mechanisms is essential.

**Research Contributions.** By decoupling MLLM training from explicit 3D input requirements, `Struct2D` promotes research into modular, scalable instruction-tuning pipelines that can generalize across environments with different sensor setups. Furthermore, our public release of `Struct2D-Set`— a large-scale spatial instruction dataset built with a principled blend of structured prompts, egocentric frames, and metadata—contributes valuable benchmarks to the broader vision-language community.

