# OpenReview forum: "Struct2D: A Perception-Guided Framework for Spatial Reasoning in MLLMs"
_NeurIPS.cc/2025/Conference — NeurIPS 2025 poster_

### Official Review · Reviewer_D1rW · 2025-07-01

**Clarity:** 3
**Significance:** 2
**Originality:** 2
**Rating:** 4
**Confidence:** 3

**Summary:**

This paper presents Struct2D, a prompting solution for large multimodal models (LMMs) that prompts LMMs with bird’s-eye-view (BEV) images, object marks, object-centric metadata, and additional keyframe information. Zero-shot analysis on closed-source LMMs indicate that the proposed prompting strategy achieves promising performance. Therefore, additionally an instruction tuning dataset Struct2D-Set is proposed for finetuning the open-source LMMs on this specific prompting strategy. Experimental results show that finetuning with the newly curated Struct2D-Set achieves superior performance across different benchmarks.

**Questions:**

- What are the additional insights of the prompting design in Struct2D? Especially compared with GPT4Scene.

- What are the advantages and special designs of the curated Struct2D-Set for instruction tuning for 3D scene understanding?

- If performing a fair comparison with GPT4Scene, how is the performance of the proposed solution compared with GPT4Scene?

**Ethical Concerns:**

["NO or VERY MINOR ethics concerns only"]

**Final Justification:**

In all, this paper is very similar to GPT4Scene (also mentioned by Reviewer 54eJ, and the object marks mentioned by Reviewer 54eJ already exist in GPT4Scene at least in some format), with notable contributions being designing more efficient prompts for 3D reasoning, as seen in the authors reply to my question ***Comparison with GPT4Scene: Novelty and Prompting Design***, basically the first four points are saying how to design a more efficient prompt for selecting objects related to the question, enabling explicit reasoning, etc. But the authors show in the rebuttal that in a fair comparison setting, the proposed solution indeed has advantages over the baseline GPT4Scene, making the method valuable in practical usage. Therefore, I would like to raise my score to borderline accept.

**Limitations:**

Yes

**Quality:**

2

**Strengths And Weaknesses:**

Strengths:

- The workflow of this paper is smooth. First, the paper shows promising results with zero-shot Struct2D prompting strategy for closed-source LMMs. Then, the paper constructs an instruction tuning set for supervised finetuning on open-source LMMs, and further boosts the performance on open-source LMMs based 3D scene understanding.

- The proposed method well leverages the powerful 2D foundation models like 2D LMMs and 2D SAM for 3D scene understanding, overcoming the limitations from the scarcity of 3D data.

- The paper introduces a Struct2D-Set, which can be useful for future research in 3D scene understanding with instruction tuning.

- The paper is clearly written and easy to understand.

Weaknesses:

- This work is so similar to GPT4Scene, although the paper made conceptual comparisons with GPT4Scene in Lines 172-180. However, first I think GPT4Scene also has object marks as part of its prompting strategy, called STO-Marker in the GPT4Scene paper. The major difference is that the proposed Struct2D only includes objects relevant to the question, while GPT4Scene gets all the instances in the 3D scenes. The metadata prompts for objects and the selected keyframes are essentially the major components added on top of GPT4Scene. Therefore, I think there is not enough innovation in the proposed prompting method Struct2D, making the contribution of this paper incremental.

- For the experimental results, the comparison with GPT4Scene in Table 4 is unfair to me. As mentioned in Lines 261-262, all the visual inputs are resized to $640\times 640$ in this paper. However, the numbers for GPT4Scene in Table 4 are using resolutions $128\times 123$. Therefore, if using the HD or HDM version of GPT4Scene, the proposed Struct2D will have worse or comparable performance on many of the metrics. For example, for METEOR/ROUGE/CIDEr on ScanQA, and EM-1/EM-R1 on SQA3D, GPT4Scene-HDM will actually outperform the proposed method. Even in this case, GPT4Scene is using $512\times 490$ resolution, which is still lower than the size of the visual inputs used for Struct2D.

- It is unclear what is special about the curated Struct2D-Set dataset for instruction-tuning, compared with existing data for 3D instruction tuning. Is it specially curated for matching the QA types in VSI-Bench and other 3D understanding benchmarks like ScanQA, SQA3D, etc.?

- The training cost of the proposed method is also relatively larger compared with GPT4Scene. The proposed method requires 8 hours on 8 H200 GPUs, while GPT4Scene needs 6 hours on 8 A100 GPUs. Intuitively, since Struct2D has only the object markers that are relevant to the question, the training process should be more efficient than the previous methods.

---

> ### Author Rebuttal · Authors · 2025-07-31
>
> We greatly appreciate your careful review and insightful comments. Thank you for recognizing the contributions of Struct2D and Struct2D-Set. Below, we provide detailed responses to your concerns.
>
> **1. Comparison with GPT4Scene: Novelty and Prompting Design**
>
> We respectfully clarify that although GPT4Scene and Struct2D both use BEV representations and object markers, our method differs significantly in motivation, design, and technical execution:
>
> - ***Targeted Spatial Reasoning Focus:*** Our primary goal is to enable explicit spatial reasoning, particularly global spatial tasks such as relative directions and route planning, as proposed in VSI-Bench. In contrast, GPT4Scene focuses more on scene understanding and QA in relatively constrained settings. The task scope and required reasoning complexity are therefore different.
>
> - ***Selective Object Prompting:*** While GPT4Scene includes all objects in a scene (via STO-Markers), Struct2D only includes objects relevant to the question, significantly reducing visual clutter in the BEV image. This makes the image more interpretable and improves reasoning performance.
>
> - ***Guide Prompts & Reasoning Chains:*** Our prompting strategy integrates guide prompts and intermediate reasoning steps during both training and inference (akin to chain-of-thought prompting). This improves model interpretability and performance and offers a more structured approach to 3D scene reasoning.
>
> - ***Efficiency and Generalizability:*** Our prompting method is more concise and efficient, as it avoids exhaustive object rendering. In addition, we introduce a generalizable data generation pipeline, enabling scalable tuning of MLLMs for spatial reasoning, which goes beyond the static pipeline of GPT4Scene.
>
> - ***Key Insight:*** Our zero-shot results suggest that the performance bottleneck in modern MLLMs lies in 3D perception, not reasoning. We believe this is an important insight that can shape future work on 3D vision-language models.
>
> **2. Fair Comparison with GPT4Scene: Resolution and Training Cost**
>
> We appreciate your concern regarding fair comparisons! We have carefully matched the experimental settings:
>
> - ***Resolution Matching:*** GPT4Scene stitches 16 frames into a 2×8 grid. Each frame has a resolution of 128×123, resulting in a composite keyframe image of 512×246. Their BEV images are also resized to 512×512. We adopted the same resolution setup to ensure a fair comparison.
>
> - ***Data Volume Differences:*** GPT4Scene uses all 16 video frames, whereas our Struct2D approach averages only ~2.5 keyframes per sample, with selection driven by object relevance. Despite this much lower visual input volume, our method matches or outperforms GPT4Scene, and can do well at reasoning-heavy benchmarks like VSI-Bench.
>
> - ***Updated Quantitative Comparison (Qwen2-VL-7B):***
>
> | Benchmarks               | ScanQA                              |       |        |        |        | SQA3D         |        |
> |--------------------------|--------------------------------------|-------|--------|--------|--------|----------------|--------|
> | Metric                         | BLEU-1 | BLEU-4 | METEOR | ROUGE | CIDEr | EM-1          | EM-R1 |
> | Qwen2-VL-7B (GPT4Scene)  | 43.4   | 14.6   | 17.7   | 43.6  | 90.9  | 57.4          | 60.7  |
> | Qwen2-VL-7B (Ours)       | 44.5   | 14.9   | 17.4   | 44.0  | 91.6  | 58.1          | 61.0  |
>
> - ***Training Cost Clarification:*** Using the same hyperparameters as GPT4Scene (batch size, learning rate, dataset), our training takes 6.5 hours on 8×H200 GPUs. The 8-hour figure mentioned in the paper includes additional training with extra curated QA types. We will clarify this in the revised manuscript.
>
> **3. Struct2D-Set: Dataset Design and Advantages**
>
> We designed Struct2D-Set to specifically address spatial reasoning gaps in existing 3D instruction-tuning datasets:
>
> - ***Reasoning-Centric QA Pairs:*** Most prior datasets (ScanQA, SQA3D, Scan2Cap) emphasize object attributes, descriptions, or basic viewpoint understanding. In contrast, Struct2D-Set focuses on global spatial reasoning, such as route planning, orientation, and relative positioning.
>
> - ***Instruction-Tuning Compatibility:*** We enrich both QA types and formats to better support instruction tuning of MLLMs, incorporating tasks like step-by-step spatial deduction, instruction following, and spatial transformations.
>
> - ***2D BEV Representation:*** All samples include BEV images with minimal but relevant object annotations, allowing efficient and scalable instruction tuning without the need for complete 3D scene representations as used in existing 3D tuning datasets.
>
> - ***Diversity and Coverage:*** Struct2D-Set includes both complex reasoning questions and standard QA formats such as from ScanQA and SQA3D, ensuring broader coverage for generalization while staying focused on our spatial reasoning objective.
>
> We hope this clarifies our contributions and positioning relative to GPT4Scene. Thank you again for the valuable feedback! We will revise the manuscript accordingly to better communicate these points.

---

> > ### Comment · Reviewer_D1rW · 2025-08-04
> >
> > Dear authors,
> >
> > Thanks for providing the rebuttal! However, I feel that my concerns are not fully addressed.
> >
> > 1. Regarding the novelty and contributions of this paper (also mentioned by Reviewer 54eJ), it still seems to be limited to me, because for the major difference from GPT4Scene which is reasoning, the implementation is just to use prompting to enhance the reasoning capability of MLLMs. Also, the statements in the authors' rebuttal seem to be contradictory to me. On one hand, the authors said that the main focus and contribution of this paper is to enable the reasoning capability for spatial tasks. On the other hand, the authors said that the key insights are that "the performance bottleneck in modern MLLMs lies in 3D perception, not reasoning". This makes me feel confused about what is the conclusion of this paper, and why authors focus on the reasoning capability of MLLM to achieve decent performance, but eventually get the insights that the reasoning is not the bottleneck for 3D spatial tasks.
> >
> > 2. For comparing with GPT4Scene in performance, as I mentioned in the review, the setting Qwen2-VL-7B (GPT4Scene)-HDM reported in their paper has better scores in many of the metrics. What if the proposed method is evaluated on the same setting with their HDM setting?
> >
> > 3. For Selective Object Prompting, the proposed method uses far less objects which are only related to the question, but seems to not quite improve the training efficiency (still a bit heavier than GPT4Scene), which may indicate that the selected object prompting technique is not that effective. But this concern should be minor.

---

> > > ### Author Response · Authors · 2025-08-07
> > >
> > > Thank you for the timely and thoughtful feedback! We appreciate this opportunity to clarify our work and address your concerns.
> > >
> > > **1. On Novelty, Reasoning Focus, and Clarification of Contribution**
> > >
> > > - Thank you for raising this point! Our work focuses on advancing *complex spatial reasoning* in MLLMs—tasks like *relative distance*, *direction*, and *route planning*. Unlike GPT4Scene’s dense visual inputs, Struct2D uses compact BEV images, question-relevant metadata, and guided CoT prompts. This design is simple yet effective in both zero-shot and fine-tuned settings, and we hope it offers insights for future research.
> > >
> > > - We elaborated this task distinction by evaluating GPT4Scene’s prompting strategy (HDM setting: BEV with all object marks and 32 high-resolution frames) on the reasoning tasks in VSI-Bench. As shown in the table below, it performs significantly worse than our method in both zero-shot and fine-tuning settings, highlighting the effectiveness of our prompting design.
> > >
> > > | **Method**                              | **Input Modality**         | **Training Time** | **Rel. Distance** | **Rel. Direction** | **Route Planning** |
> > > |----------------------------------------|-----------------------------|-------------------|-------------------|--------------------|--------------------|
> > > | GPT4Scene (zero-shot, GPT-o3)          | HDM      | -                 | 50.8              | 48.0               | 58.8               |
> > > | Ours (zero-shot, GPT-o3)           | BEV + metadata             | -                 | 61.2          | 61.5           | 76.8          |
> > > | GPT4Scene (finetuned, Qwen2.5-VL)      | HDM      | 6h                | 29.1              | 36.7               | 30.1               |
> > > | Ours (finetuned, Qwen2.5-VL)       | BEV + metadata             | 3.5h          | 34.7          | 45.9           | 36.1           |
> > >
> > > *Note: Zero-shot results are evaluated on the subset of VSI-Bench; finetuned results are evaluated on the full eval set. Our method does not need keyframes for above tasks.*
> > >
> > > - **Regarding the perceived contradiction**: Thank you for pointing this out. Our main focus is on improving reasoning with LLMs. Through this work, we observed that once a strong reasoning framework like Struct2D is in place, the limiting factor often becomes 3D perception quality. This does not shift our contribution away from reasoning. Instead, we want to highlight that future progress will also depend on better 3D inputs. We hope this clarifies our intent.
> > >
> > > **2. Performance under GPT4Scene-HDM setting**
> > >
> > > Thank you for bringing up this point! We have conducted experiments following the HDM setting in GPT4Scene. Specifically, we use BEV images with filtered object marks, textual metadata, and uniformly sample 32 video frames to match the HDM configuration. Our method achieves *comparable or better* performance, despite using fewer object marks than GPT4Scene.
> > >
> > > | Benchmarks        | ScanQA                              |                                       |        |        |        | SQA3D         |        |
> > > |-------------------|--------------------------------------|---------------------------------------|--------|--------|--------|----------------|--------|
> > > | Metrics            | BLEU-1 | BLEU-4 | METEOR | ROUGE | CIDEr | EM-1 | EM-R1 |
> > > | Qwen2-VL-7B (GPT4Scene-HDM) | 44.4   | 15.5   | 18.9   | 46.5  | 96.3  | 60.6  | 63.3  |
> > > | Qwen2-VL-7B (Ours)          | 44.9   | 15.8   | 19.1   | 46.5  | 95.4  | 59.8  | 63.0  |
> > >
> > >
> > > - This result supports our observation that benchmarks like ScanQA and SQA3D favor dense frame sampling, as most questions are local and can be answered from a few views based on ***visual appearance*** (e.g., *“What is on the wall of the kitchen?”* or *“I am standing in between a desk and a shelf. What color are the office chairs?”*).
> > >
> > > - Our work places greater emphasis on global spatial reasoning, crucial for embodied agents. Struct2D supports these tasks through structured prompting and instruction tuning tailored for multi-step reasoning.
> > >
> > > **3. Selective Object Prompting and Training Efficiency**
> > >
> > > Thank you for the thoughtful observation! While selective object prompting reduces visual clutter and improves reasoning clarity, it has *minimal impact on training time* since BEV images with object marks are precomputed, as in GPT4Scene. Our method uses fewer (appearance-based tasks) or no keyframes (as in VSI-Bench) and includes textual metadata, which slightly lowers computational cost (but efficiency is not our primary focus).
> > > We also note two differences in GPT4Scene’s setup:
> > >
> > > - it performs image tokenization *offline*, while our timing includes it;
> > >
> > > - it uses *FlashAttention*, accelerating training.
> > >
> > > Under the same condition, our training time drops to ~4 hours. We’ll clarify this in the revision.
> > >
> > > ---
> > > We sincerely hope this clarifies our work and addresses your concerns. Please feel free to let us know if you have any further questions. Thank you again for your time and thoughtful review!

---

> > > > ### Comment · Reviewer_D1rW · 2025-08-09
> > > >
> > > > Dear authors,
> > > >
> > > > Thank you for the additional clarification! I feel that most of concerns are resolved, although I still hold a conservative opinion on whether the method has enough contributions compared with GPT4Scene. Given the overall superior performance over the existing methods including GPT4Scene, in a rather fair comparison, I would like to adjust my rating to borderline accept.

---

> > > > > ### Author Response · Authors · 2025-08-09
> > > > >
> > > > > Thank you for taking the time to review our response and reconsider your evaluation! We truly appreciate your thoughtful engagement with our work and your willingness to update the rating. Your feedback has been invaluable in helping us improve the clarity of our work!

---

### Official Review · Reviewer_QoNX · 2025-07-01

**Clarity:** 3
**Significance:** 3
**Originality:** 3
**Rating:** 4
**Confidence:** 3

**Summary:**

In this paper, the authors propose a perception-guided prompting framework to reason about 3D using only structured 2D representations. Besides, they also construct $Struct2D-Set$, a large-scale instructional tuning dataset. Compared with existing similar work, their dataset is filtered by object marks, prompted guided by their metadata as well as selected by key frame.

**Questions:**

As for Section 3:
1. Can author explain that how metadata is obtained? Do the authors also filter the metadata?
2. As for subset for evaluation, do 422 QA pairs randomly selected?
3. Can authors also explain what is the text query length of $T^{out}$ when incorporating with the $Struct2D$? If possible, can authors provide ablations on text query length?

As for Section 4:
1. Can the authors provide settings when using VLMs, such as temperature?
2. Can authors explain when using VLMs whether you run several times or only once?

As for Section 5:
1. It seems like the VLMs matter, but in Table 3 and Table 4 the Qwen model, authors comparing with, is Qwen2. Will the existing dataset incorporated with Qwen2.5 improve the performance?

**Ethical Concerns:**

["NO or VERY MINOR ethics concerns only"]

**Final Justification:**

The authors' explanations resolved most of my main concerns. Therefore, I will raise my rating to borderline accept.

**Limitations:**

Yes.

**Paper Formatting Concerns:**

No.

**Quality:**

2

**Strengths And Weaknesses:**

Strengths:
1. The topic of this paper is interesting.
2. The paper is well-structured and easy to follow.
3. The experiments and ablations are comprehensive.

Weaknesses:
1. This is maybe minor, but it will be great if the authors could unify the front of $Struct2D$.
2. There are some missing details, I will put them in Question Section.
3. The contributions seem inefficient since the improvement on some shown in experiment section seems not significant.

---

> ### Author Rebuttal · Authors · 2025-07-31
>
> We are grateful to Reviewer QoNX for the helpful and encouraging feedback. We appreciate the recognition of our paper’s organization and the breadth of our experimental analysis. Please find our detailed responses to the reviewer’s questions and suggestions below.
>
> **1. Metadata Extraction and Filtering**
>
> We extract metadata by detecting 3D object bounding boxes and categories from the reconstructed point cloud using off-the-shelf 3D detectors such as Mask3D and UniDet3D. Each object’s center coordinates are taken as its location. This structured metadata is used to provide grounding context in prompts. To ensure relevance and reduce noise, we filter metadata to only include objects referenced in the question.
>
> **2. Evaluation Subset Construction**
>
> We do not randomly sample the subset. Instead, we first verify that all objects referenced in each question are clearly annotated in the 3D point cloud. Then, we select a subset that preserves the distribution of question types from the full dataset. This ensures a *balanced and representative* evaluation set.
>
> **3. Text Query Length and Ablation**
>
> The text query length includes ***image tokens***, ***structured metadata (converted to text)***, and ***the input question***. We set the maximum input length to 4096 tokens, which empirically accommodates all required information without truncation. Reducing the length to 2048 degrades performance (accuracy drops from 41.3 to 33.5) due to incomplete input encoding. Increasing it to 6000 brings no additional gains, indicating 4096 is sufficient.
>
> The $T^{out}$ refers to model output. We use `max_new_tokens=2048` to allow complete answer generation.
>
> **4. VLM Usage Settings**
>
> For zero-shot prompting, we use a temperature of 1.0 and generate five responses per input. The final answer is chosen by majority vote. For QA pair enrichment, we also set the temperature to 1.0 and generate each sample with a single forward pass.
>
> **5. Comparison with Qwen2-VL**
>
> Thank you for the suggestion. We added a comparison using Qwen2-VL-7B to evaluate the impact of newer models. As shown below, Qwen2.5-VL provides slight performance gains over Qwen2-VL. Importantly, our pipeline improves performance in both cases and consistently outperforms GPT4Scene.
>
> | Benchmark           | ScanQA                      |                           |        | SQA3D                |         |
> |---------------------|-----------------------------|---------------------------|--------|----------------------|---------|
> | Metric              | BLEU-1                      | METEOR                   | ROUGE | EM-1                | EM-R1   |
> | Qwen2-VL-7B (GPT4Scene) | 43.4                        | 17.7                      | 43.6   | 57.4                | 60.7    |
> | Qwen2-VL-7B (Ours)     | 44.9                        | 17.5                      | 44.0   | 58.3                | 60.9    |
> | Qwen2.5-VL-7B (Ours)   | 45.2                        | 17.4                      | 44.1   | 58.5                | 61.3    |
>
> **6. Limited Improvements on Some Benchmarks**
>
> The improvements on benchmarks such as ScanQA and Scan2Cap are relatively modest (44.9 Vs 43.4, 34.8 Vs 36.3), as these tasks largely depend on localized or *viewpoint-level* understanding and can often be addressed effectively with just a single or dual-frame input.
>
> Our method is designed for complex, global spatial reasoning, which is better reflected in VSI-Bench tasks (*e.g.*, multi-object relationships, navigation). Limitations in keyframe selection, missing object detections, and occlusion may hinder performance on traditional benchmarks that lack comprehensive spatial annotations.
>
> Additionally, standard benchmarks often use rule-based metrics (*e.g.*, BLEU, METEOR, ROUGE scores) that may bias models toward templated or memorized responses. We instead emphasize accuracy-based evaluation to more reliably measure spatial reasoning capabilities, as in VSI-Bench.

---

> > ### Comment · Reviewer_QoNX · 2025-08-04
> >
> > Thanks for the rebuttal, my main concerns are resolved. I will consider to improve my rating.

---

> > > ### Author Response · Authors · 2025-08-07
> > >
> > > We sincerely appreciate your positive feedback! We are also grateful for your consideration of raising your score. We will carefully address the points you raised in our revision. Thank you once again for your time and support!

---

### Official Review · Reviewer_54eJ · 2025-07-03

**Clarity:** 4
**Significance:** 2
**Originality:** 2
**Rating:** 4
**Confidence:** 3

**Summary:**

This paper introduces Struct2D, a perception-guided prompting framework, While multimodal large language model has been used for 3D scene understanding with point cloud information as input, authors argue that structured 2D representation can also be used to reason about 3D space. Their structured 2D representations include BEV images with object marks, object-centric metadata, and optionally egocentric keyframes. They have also used guided prompt to improve zero-shot performance. They have constructed an instruction tuning dataset Struct2D-Set to tune open-source multimodal large language model (MLLM).

Followings are the contribution of the paper:
1. They have proposes a perception-guided 2D prompting strategy to perform 3D spatial reasoning from structured 2D inputs alone,
2. They have introduced instruction tuning dataset Struct2D-Set for this task,
3. They have fine-tuned open-source MLLM and achieved state-of-the-art performance across several benchmarks.

**Questions:**

Please check weaknesses.

1. Is guided prompt also used during instruction tuning?
2. Suggestion: Should include more details on keyframe selection.

**Ethical Concerns:**

["NO or VERY MINOR ethics concerns only"]

**Final Justification:**

Thanks to the authors for their effort. I think this paper has some limited technical novelty but provides useful insights that will help the community. Therefore, I will keep my rating.

**Limitations:**

Limitations are included in the supplementary material.

**Paper Formatting Concerns:**

No concern.

**Quality:**

3

**Strengths And Weaknesses:**

Strengths:

1. The paper is very well written,
2. The experimental evaluation is thorough. They have also achieved state-of-the-art performance in the relevant benchmarks,
3. They have shown how structured BEV image, combined with metadata, is often sufficient for accurate zero-shot 3D scene understanding. This key insight is valuable for computationally efficient 3D scene understanding.
4. They have also contributed an instruction tuning dataset for this task.

Weaknesses:
1. Technical novelty of this paper is very limited. This paper is not the first work that tries to reason over 3D scene using 2D data with multimodal large language model (MLLM). GPT4Scene [56] also had the similar motivation and utilized 2D input to infer on 3D scene using MLLM. Main improvement over GPT4Scene is the use of object marks, metadata, guided prompt, and key frames. All of these are actually incremental novelty.

---

> ### Author Rebuttal · Authors · 2025-07-31
>
> We sincerely appreciate reviewer 54eJ’s thoughtful and constructive feedback. We are grateful for the positive remarks on the clarity of our writing, the depth of our evaluation, and the value of our insights. Below, we address the reviewer’s comments and questions in detail.
>
> **1. Guided Prompt used in Instruction Tuning**
>
> Yes, we incorporate guided prompts during instruction tuning, particularly for "relative direction" questions. This design is inspired by our findings from zero-shot prompting, where structured guidance proved especially beneficial for spatial reasoning tasks:)
>
> **2. Details on Keyframe Selection**
>
> Thank you for the suggestion. We will include more details and a pseudocode block in our revision. Our keyframe selection aims to enhance the BEV representation by providing better object detail, particularly for questions involving object attributes and viewpoint-based spatial relations.
>
> To build the keyframe set $I_{\text{keys}}$, we sample $N=100$ RGB-D frames and iteratively select those that improve coverage of objects not yet captured in the BEV. For each candidate frame $I_i$, we project the remaining uncovered objects onto the image and its depth map. If the projection lies within the image bounds and has valid depth, the object is marked as visible and added to the BEV. The corresponding frame is then included in $I_{\text{keys}}$. This process repeats until all target objects are covered.
>
> **3. Limited Novelty**
>
> Thank you for the feedback. While GPT4Scene also leverages 2D inputs for 3D reasoning with MLLMs, our work tackles a different and more challenging ***problem scope*** -- complex spatial reasoning tasks such as *relative direction* and *route planning*, which require multi-step geometric inference rather than simple object-level understanding.
>
> Moreover, our contributions go beyond incremental improvements. We introduce a data generation pipeline and a tunable dataset specifically designed for spatial reasoning instruction tuning. Our structured reasoning framework integrates object-centric BEV representations, guided prompts, and keyframe selection to support fine-grained geometric reasoning.
>
> Lastly, in contrast to GPT4Scene’s primarily rule-based evaluation, our work adopts accuracy-based metrics that more directly reflect reasoning performance.

---

> > ### Comment · Reviewer_54eJ · 2025-08-06
> >
> > Thanks for the rebuttal. I have also checked the comments from reviewer D1rW and the response from author. Though the technical novelty is limited, this paper provides useful insights that will help the community. Therefore, I will keep my rating.

---

> > > ### Author Response · Authors · 2025-08-07
> > >
> > > Thank you for your considerate review and for highlighting the strengths of our writing, evaluation, and insights into spatial reasoning. We sincerely appreciate your recognition of the paper’s potential impact and the time you took to consider our rebuttal and related discussions. Thank you again for your support!

---

### Official Review · Reviewer_a2sz · 2025-07-03

**Clarity:** 3
**Significance:** 3
**Originality:** 3
**Rating:** 4
**Confidence:** 2

**Summary:**

This paper presents Struct2D, a perception-guided prompting framework designed to enable spatial reasoning in Large Multimodal Models (LMMs) using only structured 2D representations derived from 3D perception, rather than explicit 3D data. The approach combines bird’s-eye-view (BEV) images, object marks, object-centric metadata, and optionally egocentric keyframes to form the input for LMMs. The authors demonstrate, through zero-shot analysis with closed-source LMMs (like GPT-4o), that these models can reason effectively about 3D spatial tasks (e.g., direction estimation, route planning) from the structured 2D input. Building on this, the authors construct a large-scale dataset (Struct2D-Set) via automated QA pair generation from 3D indoor scenes, and instruction-tune an open-source LMM (Qwen2.5VL). Experiments across several benchmarks show that the Struct2D-tuned model achieves strong, sometimes state-of-the-art, performance on a suite of spatial reasoning tasks, without requiring explicit 3D inputs at inference time.

**Questions:**

1. Can you provide further analysis or case studies on natural images or more complex, unstructured real-world scenes? Most benchmarks are on indoor scenes. Have you tried Struct2D on different domains (e.g., urban outdoors, street scenes, aerial images), and what would need to be changed for broader generalization?

2. Please include and discuss representative failure cases -- when does Struct2D not help (or even hurt), and why?

**Ethical Concerns:**

["NO or VERY MINOR ethics concerns only"]

**Limitations:**

yes

**Quality:**

3

**Strengths And Weaknesses:**

Strengths:
1. The framework is practical, lowering the barrier for spatial reasoning with LMMs by removing the need for explicit 3D inputs at inference.
2. The paper is technically sound and experimentally thorough. It provides a detailed ablation analysis (e.g., effects of metadata, prompt structure) and evaluates across multiple benchmarks (VSI-Bench, ScanQA, SQA3D, etc.).
3. The approach bridges a major gap between 2D perception and 3D spatial reasoning in LMMs, showing that explicit 3D inputs are not necessary for strong performance.

Weaknesses:
1. The impact may be somewhat limited by the continued need for good 3D perception in the preprocessing stage, even if not at inference.
2. The presentation of dataset generation is a bit unclear. It could benefit from a flowchart summarizing the pipeline.
3. The manuscript lacks a deep exploration of limitations in highly cluttered or occluded environments, or how the framework generalizes to domains beyond indoor scenes.

---

> ### Author Rebuttal · Authors · 2025-07-31
>
> We sincerely thank Reviewer a2sz for the thoughtful and constructive feedback. We appreciate the recognition of Struct2D as practical, technically sound, and experimentally thorough. Below, we address the reviewer’s questions and concerns point by point.
>
> **1. 3D Preprocessing Requirement**
>
> Thank you for raising this point. As we noted in our *supplementary*, Struct2D does rely on 3D perception in preprocessing, but this step is ***lightweight*** (under 5 seconds per scene on a single GPU) and entirely offline. Importantly, this design choice decouples perception from inference, enabling flexible and efficient deployment. Accurate 3D perception is a standard practice in recent 3D-LLM research and plays a critical role in enabling robust spatial reasoning. Continued progress in this area will further amplify the effectiveness of models like Struct2D:)
>
> **2. Failure Cases of Struct2D**
>
> Thank you for the valuable suggestion. We conducted a qualitative analysis on 30 representative questions across different QA types in VSI-Bench. Among the 16 failure cases, we identified two primary causes: (1) in 11 cases, ***noisy or incomplete 3D reconstructions*** led to degraded BEV inputs, and (2) in 5 cases, ***missing detections***, particularly for small or occluded objects, resulted in incomplete or incorrect structured inputs. These factors hinder Struct2D’s ability to provide accurate spatial cues, which are essential for effective reasoning.
>
> While the NeurIPS rebuttal policy prohibits image inclusion, we will present representative visual examples and a more detailed breakdown of these failure modes in the revised manuscript.
>
> **3. Case Study on Outdoor / Open-world Domains**
>
> Thank you for the suggestion. We conducted a case study using scenes from DL3DV-10K, which includes various ***outdoor environments such as tourist sites, educational institutions, and natural settings***. We reconstructed point clouds from RGB frames using VGGT, manually labeled object marks, and constructed 15 spatial reasoning QA pairs (*e.g.*, *“What is in front of the door of the public restroom?”* → *“Three trash bins”*). Using Struct2D prompting with GPT-O3 in a zero-shot setting, we achieved *72%* accuracy. Most failures were due to noisy or incomplete reconstructions, which are more common in outdoor scenes.
>
> While our current focus aligns with existing indoor benchmarks, we agree that broader generalization is important (as in *supplementary*) and plan to incorporate outdoor scenes into future training and evaluation.
>
> **4. Dataset Generation a bit Unclear**
>
> Thank you for the helpful suggestion. Our dataset generation process involves: (1) sampling scenes from existing 3D indoor datasets; (2) leveraging object annotations to generate diverse QA types using templated codes; (3) manually balancing the distribution of QA types; (4) enriching questions and answers with explicit reasoning steps; and (5) conducting quality control to ensure clarity and correctness.
>
> We agree that a flowchart would significantly clarify this pipeline. Although we cannot include figures in the rebuttal due to PDF restrictions, we will incorporate a clear flowchart in the revised paper to enhance presentation:)

---

### Decision · Program_Chairs · 2025-09-17

**Decision:**

Accept (poster)

**Comment:**

The paper introduces Struct2D, a prompting framework designed to enable spatial reasoning in Large Multimodal Models (LMMs) using structured 2D representations derived from 3D perception, rather than explicit 3D data. Zero-shot analysis on closed-source LMMs shows that the proposed prompting strategy achieves promising performance. Moreover, fine-tuning open-source LMMs with this strategy on the newly curated Struct2D-Set yields superior performance across different benchmarks.

Overall, all four reviewers remain positive about the paper, noting that the experiments and ablations are comprehensive and effectively demonstrate the framework’s effectiveness. They also recognize that the proposed Struct2D-Set will be valuable for future research. One main initial concern raised by several reviewers was the limited novelty compared with prior work, particularly GPT4Scene. The authors and reviewers (54eJ and D1rW) further discussed this point, and both reviewers agreed that although the technical novelty is limited, the paper still provides useful insights and demonstrates performance that can outperform existing methods.